# Signatures of T cell immunity revealed using sequence similarity with TCRDivER algorithm

Milena Vujović [1✉], Paolo Marcatili[1✉], Benny Chain [2✉], Joseph Kaplinsky [3✉] & Thomas Lars Andresen [1✉]

Changes in the T cell receptor (TCR) repertoires have become important markers for monitoring disease or therapy progression. With the rise of immunotherapy usage in cancer, infectious and autoimmune disease, accurate assessment and comparison of the "state" of the TCR repertoire has become paramount. One important driver of change within the repertoire is T cell proliferation following immunisation. A way of monitoring this is by investigating large clones of individual T cells believed to bind epitopes connected to the disease. However, as a single target can be bound by many different TCRs, monitoring individual clones cannot fully account for T cell cross-reactivity. Moreover, T cells responding to the same target often exhibit higher sequence similarity, which highlights the importance of accounting for TCR similarity within the repertoire. This complexity of binding relationships between a TCR and its target convolutes comparison of immune responses between individuals or comparisons of TCR repertoires at different timepoints. Here we propose TCRDivER algorithm (T cell Receptor Diversity Estimates for Repertoires), a global method of T cell repertoire comparison using diversity profiles sensitive to both clone size and sequence similarity. This approach allowed for distinction between spleen TCR repertoires of immunised and non-immunised mice, showing the need for including both facets of repertoire changes simultaneously. The analysis revealed biologically interpretable relationships between sequence similarity and clonality. These aid in understanding differences and separation of repertoires stemming from different biological context. With the rise of availability of sequencing data we expect our tool to find broad usage in clinical and research applications.

[1] DTU HealthTech, Department of Health Technology, Technical University of Denmark, Kgs. Lyngby, Denmark. [2] UCL Division of Infection and Immunity, University College London, London, UK. [3] Ludwig Institute for Cancer Research Ltd, University of Oxford, Nuffield Department of Medicine, Oxford, UK. ✉email: milvu@dtu.dk; pamar@dtu.dk; b.chain@ucl.ac.uk; joseph.kaplinsky@ludwig.ox.ac.uk; tlan@dtu.dk

An individuals T cell repertoire changes throughout a lifetime as a result of foreign- and self-stimulation. The T cell compartment of adaptive immunity plays a crucial role in cancer immunity, auto-immune and infectious diseases. Adaptive immune responses as a whole draw on diverse T cell receptors (TCRs). This vast pool of TCRs is generated through the imprecise stochastic process of V(D)J recombination giving rise to $10^{20}$ (or more) possible TCR combinations[1,2]. The number of possible generated TCRs is much larger than those estimated to be present within any individual T cell repertoire[3–5]. T cells activate and proliferate upon epitope-specific contact, thereby increasing their clone size. Due to the phenomenon of epitope spreading, T cells can diversify their antigen-specific response by reacting to non-dominant epitopes present on the antigen, in addition to the main dominant-epitope driven response[6,7]. Moreover, T cells are cross-reactive meaning that a single TCR can bind multiple epitope targets. However, T cells responding to the same antigen might not proliferate and increase their clone size in the same manner thereby adding complexity to the T cell-epitope response. Most importantly, this binding promiscuity ensures broad epitope recognition responses, despite the limited number of unique TCRs within each repertoire[8–10]. On the other hand, a single epitope can be bound by multiple different TCRs. It has been shown that T cells responding to the same epitope share more sequence similarity[11]. This complexity of composition of TCR repertoires makes it difficult to compare and stratify individuals based on immune status or to even establish a healthy baseline.

Initial experimental approaches, such as spectratyping[12–15] and flow cytometry[16,17] aimed to reveal oligoclonal expansions of T cells by tracking clonal sizes of $CDR_{\beta}3$s with the same length with the aim of capturing immune responses. However, T cell expansions in an organism are not exclusive to encounters with disease-associated antigens and these methods provided no insight into TCR similarity. Recent advances in high throughput sequencing (HTS) now allow for characterisation of adaptive immune receptors in increasing depth and with improved quantitation. HTS methods supply information on both clone sizes and sequence relatedness. However, this development has given rise to the need for summary measures to interpret the data generated by such experiments. Several methods have emerged to fulfil the demand to stratify repertoires either by the TCR antigen specificities[11,18,19] or by finding characteristics of TCR sequences[20–24]. Still, many of the employed methods aim to uncover epitope similarity without simultaneously examining T cell clone sizes, or vice versa. We hypothesised that measures that capture the global repertoire structure by incorporating both characteristics of the adaptive immune response could potentially be used to stratify patients for disease outcome or therapy. Providing the complexity of the immune response is encoded both in sequence similarity and TCR clone sizes, such an analysis could allow for applications in immune monitoring e.g., following immunisation, therapy at different points in time or disease progression by comparison of "snapshots" of the repertoire state. Thereby, transcending the notion of "public TCRs" into "public repertoire structures" responsible for therapeutic outcome.

To address this hypothesis we sought to further develop a popular approach in characterisation of repertoires through measures of diversity. They have been widely used in evaluation of therapy and disease effects[23,25–29] or attempts at repertoire classification and diagnosis[4,30,31]. Yet, there are a variety of ways in which the intuitive idea of diversity can be formalised. In the naive sense, diversity is estimated based on the number of and clone sizes of unique TCRs in a repertoire. Commonly used diversity estimates are richness (number of different TCR clones), clonality (number of large clones) and diversity indices such as Shannon entropy[32], Simpson[33], Gini-Simpson[34] and Berger-Parker[35] index. Different diversity indices will weight large clones differently, thereby imposing a threshold on the clone frequencies within the repertoire. Thus, counting unique clones by richness will give rare or small clones the same weight as large clones. Entropy, will give more weight to large clones than to small or rare clones. No single index will capture all information about the clone size distribution. Notably, this ambiguity has led to no clear consensus which diversity index should be applied in practical cases of interpreting immune diversity[36,37]. The approach of using individual diversity indices provides no repertoire characteristics truly independent of sample size[3]. Moreover, it can lead to erroneous conclusions on ordering repertoires based on the choice of diversity index (Fig. 1a). And finally, measures of diversity should not only rely on clone counts, but should also account for sequence similarity of receptors.

The first problem of debatable usage of individual indices, can be surmounted by estimating them simultaneously in a single expression of diversity: the diversity of order $q$, $D(q)$, which subsumes most of the commonly used indices[34,38]. Accounting for the distribution of clone sizes, diversity can be estimated in the form of "diversity profiles"[37,39,40]. Such profiles define "effective numbers" of receptors when viewed at different resolutions, making use of a single parameter ($q$) to systematically shift focus from counting each unique clone to giving weight only to the largest clone in a repertoire (Fig. 1a, c). The use of diversity profiles gives insight into T cell clone size distributions, as the relationship between diversities calculated at different clonality weights $q$ can be correlated to the ratio of large to small clones within a repertoire[41]. This approach has been previously implemented for one B- and three T cell repertoires in the work of Greiff et al.[39]. Keeping in mind that the study focused solely on clonal frequency, the authors report remarkable separation based on immunological status, in 3 out of 4 immune repertoire datasets. However, as naive diversity estimates only take clone frequencies into account they are not sensitive to minor polyclonal expansions of TCRs reacting to the same antigen, which can theoretically mount a unified front of antigen-specific similar T cells.

The second problem, of incorporating sequence similarity in diversity estimates, has been less thoroughly explored. One approach is to count clusters of similar receptors[19]. Another approach is to use an effective number with sensitivity to sequence similarity[42]. The authors find that "similarity redefines the diversity of complex systems" and it is lower than naive diversity. This indicates that there are cohorts of similar TCRs in the repertoires and it is not excluded that they might have the same biological function e.g., bind similar epitopes. These approaches suffer from a similar limitation as use of a single diversity index in that they adopt either a single arbitrary cutoff or a single sequence similarity distance in their definition of effective number e.g., a single similarity corrected diversity index. Here we make use of approach used by Leinster and Cobbold[41] in ecology to explore 2 dimensional profiles of effective numbers. Our approach of using similarity scaled diversity $D(q,\lambda)$ allows for simultaneous characterisation of the clonal distributions and similarity of receptor repertoires (Fig. 1b). Instead of depending on a single parameter, $q$, our profiles depend on two parameters, $q$ and $\lambda$. As in conventional diversity profiles, the $q$ parameter probes the structure of the clone size distribution. The $\lambda$ parameter plays an analogous role for sequence similarity (Fig. 1d). As $\lambda$ varies from infinity down to zero the effective diversity gradually merges together more and more similar sequences. Incorporating this additional aspect to the diversity estimation allows us not only to probe the clone size distribution, but also TCR similarity which may provide information on repertoire

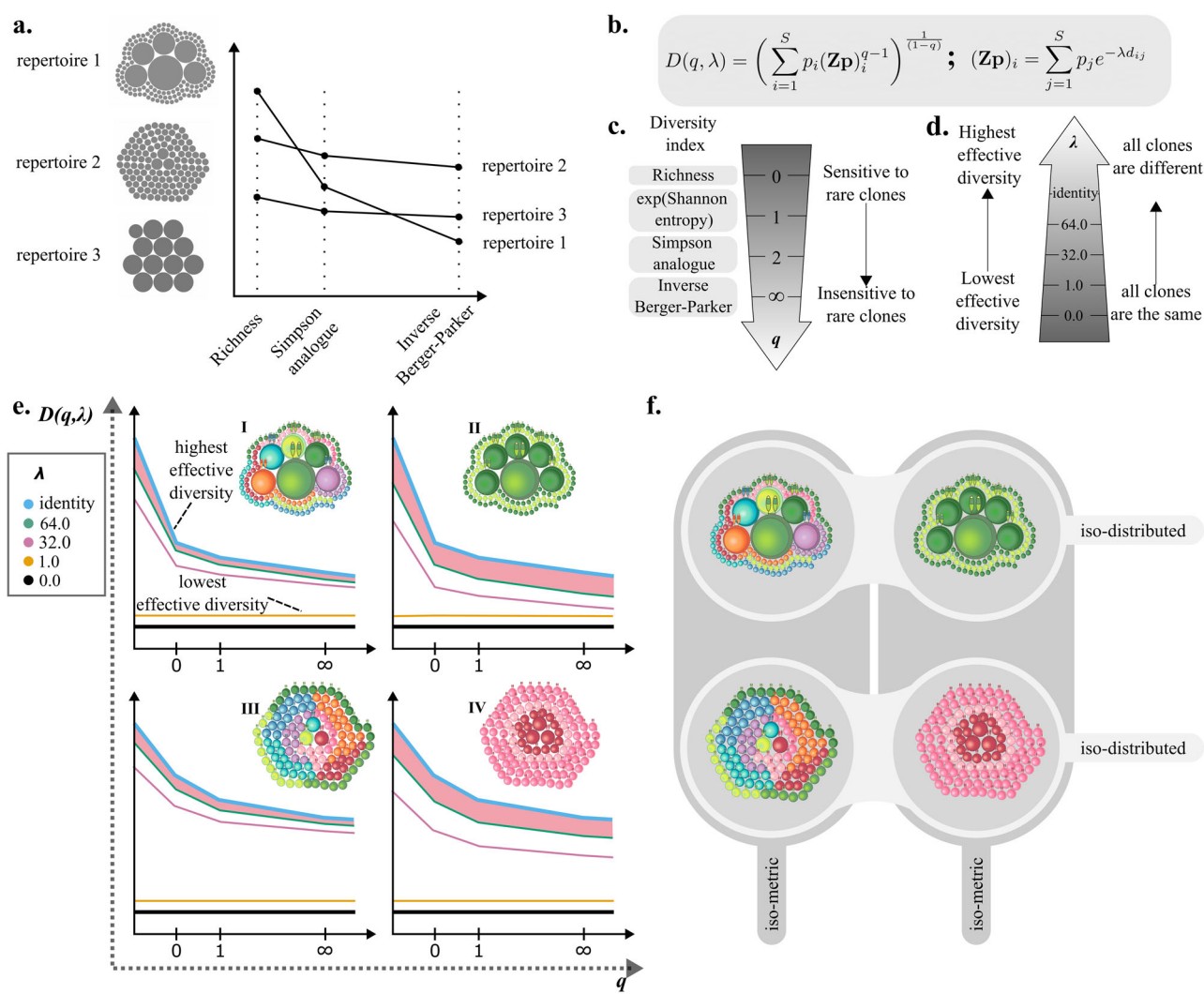

**Fig. 1 An overview of diversity estimates and their application on T cell receptor repertoires. a** Naive diversity indices for three model repertoires. The repertoires contain the same total number of T cells. Each T cell clone within the repertoire is presented as a grey circle and the size of each circle corresponds to its relative frequency. The number of unique clones falls from repertoire 1 through to repertoire 3. As different diversity indices are applied, the ordering of repertoires changes. **b** Formula for calculating similarity scaled diversity $D(q, \lambda)$. Here $p_i$ is the fraction of cells in clone $i$, Z is a similarity kernel between clones, $q$ controls sensitivity to clone size, and $\lambda$ controls sensitivity to sequence similarity. **c** Relationship of some commonly used diversity indices to the naive diversity of order $q$. **d** Effect of introducing the $\lambda$ distance scaling in the diversity calculation. As $\lambda$ increases the distance between clones increases until at $\lambda = \infty$ clones have no similarity i.e. the similarity kernel **Z** is the identity. **e** Schematic representation of diversity profiles of four model repertoires are shown along the repertoires in the top right corner of each diversity profile. Repertoire composition is schematically represented with T cells of varying size and colour. Analogously to A. the size of individual cells corresponds to the clone frequency within the repertoire. More similar colouring indicates higher T cell sequence similarity in the repertoire. Individual curves correspond to diversity profiles calculated at different $\lambda$ thresholds. One of the parameters explored with TCRDivER is highlighted in red - area between lambda curves for higher values of $\lambda$. This feature is correlated with the overall weighted similarity of the repertoire, where repertoires with higher TCR similarity exhibit a larger overall drop in diversity when similarity is included giving rise to a larger area between the curves. **f** Further explanation of structure differences between the four model repertoires in **e**. Repertoires sharing the same clone distributions are shown in rows (I = II and III = IV). Repertoires sharing same similarity relationships between T cell clones are shown in columns (I = III and II = IV).

convergence through expansion of similar clones. This concept can be visualised in changes of diversity profiles of four mock repertoires linked to varying sequence similarity and clonal distributions which are represented in Fig. 1e, f.

In this study, we showcase a new tool for estimating and comparing TCR repertoire diversity using similarity sensitive diversity estimates: TCRDivER. TCRDivER compares similarity scaled diversity of snapshots of repertoires stemming from different immunological settings, be it different treatments, diseases, time-points or their combinations. We would like to note that even though the algorithm can be used to compare repertoires from

different timepoints, it at no point takes in longitudinal data as input. We apply TCRDivER to previously published murine TCR$\beta$ sequence data from CD4$^+$ T cells following immunization[21]. We show that TCRDivER, by simultaneously probing clonal expansion and sequence similarities reveals novel TCR repertoire traits. Using features of the similarity scaled diversity profile we detect differences in response to immunisation protocols at all sampling times, indicating unique features arise within repertoires can be detected as early as 5 days and persist for several months.

Notably we find strong nonlinear correlations between features of the similarity scaled diversity profiles, including features which

characterise the average distance between sequences or the balance between large and small clones. This correlations indicate biological constraints on repertoire development which link together clone size with sequence similarity. The nonlinear shape of the correlations reveal that these features can exist in multiple discrete equilibria. Most importantly upon randomisation of T cell repertoires these features lose their nonlinear correlations, which indicates these are a natural consequence of expansions of biological repertoires.

We validate our finding that TCR repertoires reside in a non-linear space on an independent dataset of human bulk TCR sequences extracted from non-small-cell lung cancer (NSCLC) patients following CTLA-4 blockade treatment[43]. We show that by estimating repertoire diversity with TCRDivER we can unearth information which might allow us to understand more subtle differences between repertoires, stratify them and ultimately guide therapy regimes.

## Results

### TCRDivER is a tool for estimating T cell repertoire diversity based on clonality and TCR similarity.
TCRDivER is an algorithm for analysis of TCR repertoires that accepts a list of $CDR_\beta3$ amino acid sequences with their corresponding counts or frequencies within the repertoire in the form of a .tsv file (Fig. 2a). Each repertoire is processed individually giving out a .tsv file containing all the values of similarity scaled diversity $D(q, \lambda)$ for a list of chosen $\lambda$ and $q$. Individual repertoire analysis begins with $CDR_\beta3$ sequence filtering and subsampling in order to reduce computational cost and keep only sequences of interest (Fig. 2b). The filtering consists of removing all "out-of-frame" sequences and those containing "stop" codons, leaving only "In" frame $CDR_\beta3$ sequences. The filtered repertoire is then sampled for non-unique $CDR_\beta3$s randomly based on the frequency (count) distribution in the original repertoire in order to preserve the $CDR_\beta3$ frequency distribution. The subsampled $CDR_\beta3$s are then collapsed to form a list of unique $CDR_\beta3$s with their respective counts (frequencies) within the subsampled repertoire. The filtered and subsampled repertoire information on individual $CDR_\beta3$ sequences is used to calculate the naive diversity values across varying parameter $q$ (Fig. 2d). Any calculation involving an all-against-all comparison will inevitably scale with the square of the number sequences. In order to compare similarity of a large number of sequences in a memory and computational cost-efficient way, the $CDR_\beta3$ sequences comprising the now reduced repertoire, are split into ordered smaller portions or "chunks" for the distance matrix calculation (Fig. 2c and Supplementary Fig. 1). The "chunks" are formed by dividing the list of all $CDR_\beta3$s from the filtered and subsampled repertoire. Each portion of the list ("chunk") is then pairwise compared with the full list of $CDR_\beta3$s from the filtered and subsampled repertoire. These portions of the distance matrix contain n rows (n being the chunk length, default value = 100) and S columns (S being the total number of $CDR_\beta3$ sequences in the filtered and subsampled repertoire and are outputed in a .tsv format. If concatenated, these portions of the distance matrix form the full $CDR_\beta3$ distance matrix and its diagonal is equal to 0.

These files along with the frequency information for each $CDR_\beta3$ sequence are used as inputs for calculating values of true similarity scaled diversity values across a range of $\lambda$s and $q$s (Fig. 2e). Combined, all the diversity values can then be used to generate diversity profiles for each repertoire (Fig. 2f). The first part of the algorithm is intended for use in commonly available computer clusters.

Because TCRDivER is parallelisable, with the 50,000 sequences per sample analysed and distributed over 8 cores of an Intel Xeon

Gold 6126 2.60 GHz processor, each repertoire computation took under 6 h. To support further investigation and interpretation of TCRDivER results a jupyter notebook is provided. Diversity profiles can be visualised, as well combined for multiple repertoire analysis such as stratification and feature tracking analysis. The whole algorithm is outlined in Fig. 2. Full TCRDivER code is available for download at: https://github.com/sciencisto/TCRDivER.

To assess the capabilities of TCRDivER we analysed the frequency distribution and similarity of $CDR_\beta3$ sequences in TCR repertoires previously published in[21]. Briefly, the dataset consists of CD4$^+$ T cell repertoires harvested from murine spleens following immunisation with Complete Freund's Adjuvant (CFA) with or without the addition of Ovalbumin (OVA) antigen. The T cells were harvested post immunisation at three timepoints: early (days 5 and 14) and late (day 60). Additionally, we have analysed untreated mouse repertoires from the same study. We used TCRDivER to calculate diversities $D(q, \lambda)$, with varying orders of $q$ and $\lambda$. From these we constructed diversity profiles (divPs), which we present as graphs of the natural logarithm of diversity versus the varying order of q for each lambda (Supplementary Note 2.2). Each sample was sampled for 50,000 sequences reduce computational cost and mitigate the effects of sequencing depth. The constructed diversity profiles provide a graphically intuitive way to capture the shape of a repertoire. We highlight some of the features of diversity profiles bellow in order to develop an understanding of how they connect to structural and immunological characteristics of TCR repertoires.

### TCRDivER reveals unique TCR repertoire features.
In our framework the naive diversity profile corresponds to the case in which the receptor of each T cell clone is considered totally distinct, with no consideration of similarity to other clones i.e. the highest effective diversity. In reality there will be some degree of functional overlap between clones i.e., similarity, which will reduce the functional diversity below the naive value. The naive diversity ($\lambda = \infty$) is therefore a base case of maximal diversity. At the opposite extreme, $\lambda = 0$, all clones are considered functionally identical. Biologically, this would correspond to a fictional scenario where all the TCRs have no antigen-specific binding preference and would bind promiscuously to all peptide-MHC complexes, because all T cells would behave exactly the same in this scenario. In this case the functional diversity of a repertoire is therefore minimal and equals one. The parameter $\lambda$ interpolates between these two extreme cases and as the intermediate profiles in each sample correspond to intermediate values of $\lambda$. An analogy could be made with a microscope focus where the $\lambda$ parameter is able to pass through different depths of similarity within the repertoire.

We begin our account by highlighting features of the naive diversity, i.e., the upper bounding curves, in each sample. We plotted naive diversity profiles from all samples together showing that crossings in the range $0 \leqslant q \leqslant 2$ are common events (Supplementary Fig. 3). This confirmed in our dataset that the ranking of repertoires based on a single value of $q$ would indeed depend strongly on the chosen index (similar to what is shown in example Fig. 1a). We concluded that the previously mentioned justification for analysing profiles across a range of orders $q$ is not merely theoretical.

The highest value of the naive diversity at $q = 0$ gives the number of unique TCR sequences observed in the sample of 50,000 sequences. At $q = \infty$ we read off the effective number of clones in the repertoire if it consisted only of the largest clones. The rate of fall of naive diversity as $q$ rises therefore encodes information about the balance between larger and smaller clones.

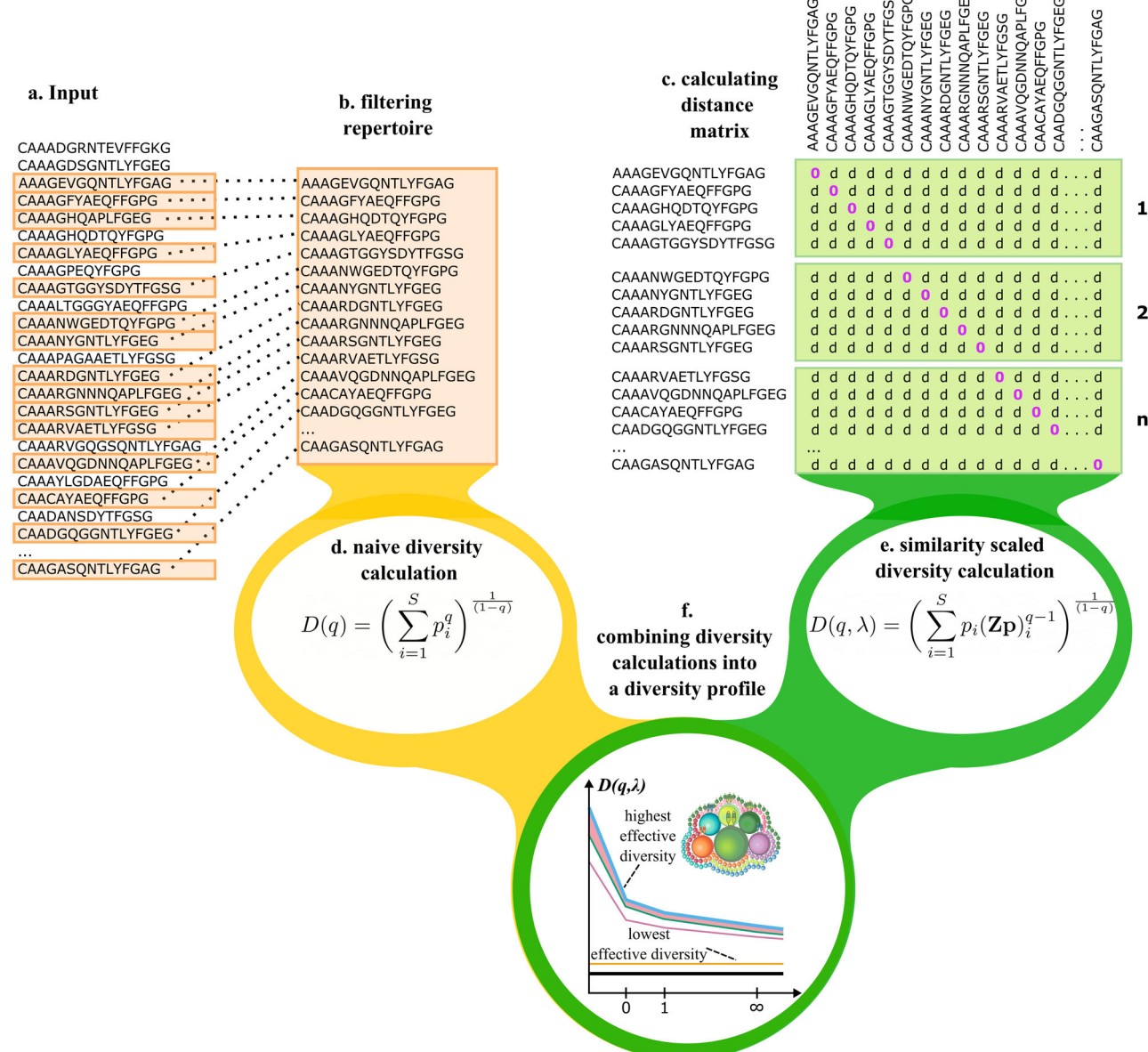

**Fig. 2 An overview of the TCRDivER algorithm. a** *Input.* The algorithm takes as input of CDR$_\beta$3 clone sequences with their clone sizes expressed as count or frequency. **b** *Filtering.* The input sample is filtered to contain only "In" frame CDR$_\beta$3 regions. Then the repertoire is sampled for CDR$_\beta$3s randomly, based on the frequency (count) distribution in the original repertoire. The default subsampling size is 50,000 CDR$_\beta$3 sequences. Meaning that 50000 non unique CDR$_\beta$3s are subsampled and counted to form the filtered and subsampled repertoire with ≤ 50,000 unique CDR$_\beta$3s with their corresponding counts within the 50,000 subsample. The CDR$_\beta$3 sequence counts in the subsampled repertoire are transformed into frequencies, so that the final output is a list of unique CDR$_\beta$3s with their respective frequencies summing up to 1 within the repertoire. **c** *Calculating distance matrix* The filtered and downsampled repertoire is provided as input for calculating the distance matrix. This step is split up so the original list of unique CDR$_\beta$3 sequences is divided to sublists of equal length which are in turn pairwise compared against the whole list of CDR$_\beta$3s using the BLOSUM45 alignment score (see Supplementary Fig. 1). The output is a set of files which contain portions of the distance matrix. When concatenated they form the complete distance matrix. In the graphical representation of the distance matrix d denotes the calculated distance between two pairs of CDR$_\beta$s. **d** *Calculating naive diversity* The script takes in only the filtered and subsampled list of CDR$_\beta$3s and their respective frequencies, since $D(q)$ is not dependent on CDR$_\beta$3 sequence distances. The output of this calculation is a .tsv file containing values of diversity at different values of $q$ (Calculating naive diversity). **e** *Calculating similarity scaled diversity* The script takes in both the list of CDR$_\beta$3s with their frequencies and the distance matrix "chunk" files. The calculation is done in a parallel fashion to reduce computational time. As output a .tsv file is given containing the values of calculated diversity at different values of $q$ and $\lambda$ (Calculating similarity-scaled diversity). **f** *Combining diversity calculations* The final step is joining the two diversity calculations in a complete overview of the diversity. Two previously obtained .tsv files with calculated diversity values are combined into one file containing all calculated values of diversity. Downstream this file is used for constructing the diversity profiles and further statistical analysis.

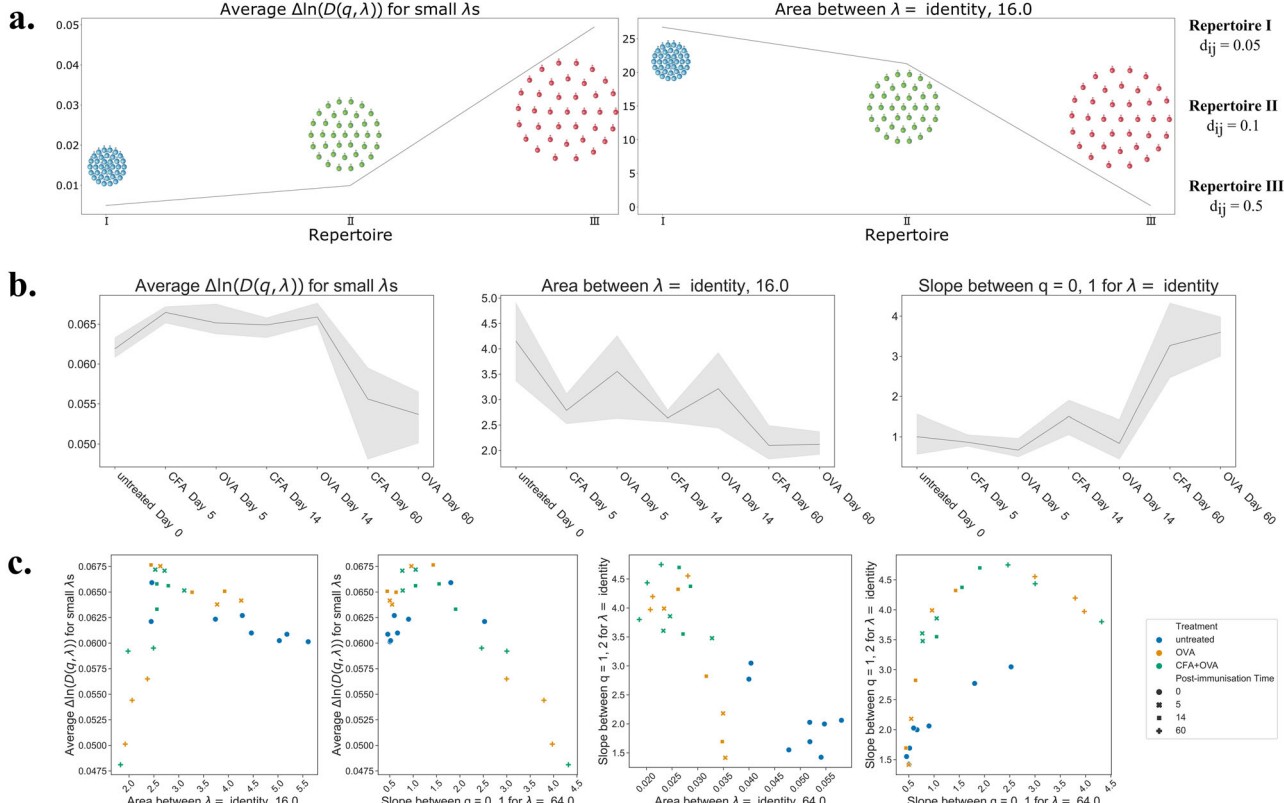

**Fig. 3 T cell receptor repertoire features derived from diversity profiles. a** Effect of changing $CDR_\beta 3$ distance in three mock repertoires on two repertoire features (average $\Delta \ln(D(q,\lambda))$ for small $\lambda$s and calculated area between $\lambda$ identity and 16). The repertoires have been generated to contain a uniform distribution of 100 $CDR_\beta 3$ clones with the $CDR_\beta 3$ distance increasing from repertoire I to III. The relationship between these two features in this simplified example is inverse. The average $\Delta \ln(D(q,\lambda))$ for small $\lambda$s is approximately linearly depended on the $CDR_\beta 3$ distance in the repertoire (Supplementary Note 1.6). **b** Trends of three features extracted from divPs are shown versus the treatment regime and timepoints ending with the latest timepoint. The features are, from left to right: average $\Delta \ln(D(q,\lambda))$ for small $\lambda$s, between curves of $\lambda =$ identity and 16.0 and slope of $q = 0 \rightarrow 1$ for value of $\lambda$ identity. The line connects the mean values of the features for all samples within a group and the shaded area represents the confidence interval. **c** Graphs showing relationships between some of the divP features. From left to right: average $\Delta \ln(D(q,\lambda))$ for small $\lambda$s is shown versus the area between curves of $\lambda =$ identity and 16.0; average $\Delta \ln(D(q,\lambda))$ for small $\lambda$s is shown versus the slope of $q = 0 \rightarrow 1$ for value of $\lambda$ 64.0; slope of $q = 1 \rightarrow 2$ for value of $\lambda$ identity (i.e. naive diversity) is shown versus the area between curves of $\lambda =$ identity and 64.0; slope of $q = 1 \rightarrow 2$ for value of $\lambda$ identity (i.e. naive diversity) is shown versus the slope of $q = 0 \rightarrow 1$ for value of $\lambda$ 64.0.

To keep in line with the microscope analogy, scaling through parameter $q$ allows us to zoom in on different resolutions of clone sizes within the repertoire. To characterise this we derived an expression for the gradient of the naive diversity at $q = 1$ and found that it is proportional to the variance of the clone size distribution i.e., the ratio of rare to common T cell clones in a repertoire (Supplementary Note 1.2).

Notably, when examining the diversity profiles shown in Supplementary Note 2.2, a sharper slope can be seen in the curves from repertoires that have been immunised compared to the untreated ones, especially for later time points. This is quantified in Fig. 3b (right side) by plotting the slope between $q = 0$ and $q = 1$ for each treatment group. The increasing value of the slope is indicative of an increased clonal expansion at later timepoints. The impact of clonal expansion in reducing diversity is seen to be mild at earlier time points after vaccination (5 and 14 days) and more marked at the day 60 time point.

In order to explore the effects of similarity scaling on measures of diversity, we investigate in depth the features of similarity sensitive profiles with values of $\lambda < \infty$. The first feature of the similarity sensitive profiles occurs near the flat curve for $\lambda = 0$ at the bottom of the profile. We observed that for small values of $\lambda$ the curves are approximately flat and are evenly spaced on a log scale in all diversity profiles, as shown in Supplementary Fig. 2

and Supplementary Note 1.8. To further investigate the biological importance of this we derived an expression for the diversity around small $\lambda$ using perturbation theory (Supplementary Note 1.6). Interestingly, the spacing showed to be proportional to the mean distance between sequences in the repertoire. In particular, the spacing of profiles, which we denote $\Delta \ln(D(q,\lambda))$ at small $\lambda$, is dependent solely on the distance and frequency of $CDR_\beta 3$s, and not on the weight $q$. We generated a simplified example of three mock repertoires (denoted I, II and III) which contain the same number of unique $CDR_\beta 3$ sequences but their relative distance within the repertoire changes. By increasing the distance between individual $CDR_\beta 3$s i.e., by reducing similarity from repertoire I to III the $\Delta \ln(D(q,\lambda))$ at small $\lambda$ increases showing the linear relationship of distance on this feature when the repertoire contains a uniform distribution of frequencies (Fig. 3a). It is important to keep in mind that this measure naturally integrates increases in similarity from both expansion of particular clones and selection of clones with similar sequence. We are therefore able to use these spacings to gain biological insight in to repertoire structure following immunisation as a measure of overall repertoire similarity. The spacing of profiles, $\Delta \ln(D(q,\lambda))$ at small $\lambda$, is presented for different treatment groups in Fig. 3b. We concluded that while there may be a small rise in spacing at early time points after vaccination (5 and 14 days),

there is a distinct decline of around 15% at day 60 indicating an increase of $CDR_\beta 3$ similarity at later time points.

The second feature is the rate at which diversity falls as $\lambda$ falls from $\lambda = \infty$ to lower values, where $\lambda = \infty$ characterises the naive profile. Unlike the case at small $\lambda$, the value of $\Delta ln(D(q,\lambda))$ at large $\lambda$ is no longer independent of $q$. It is still a measure of repertoire similarity, but heavily influenced by the clone size distribution weight $q$. Keep in mind that $\lambda = \infty$ corresponds to no effective clustering of similar sequences, that is all sequences are regarded as unique and dissimilar. Then large values of $\lambda$ correspond to just a small amount of effective clustering and counting together only the most similar clones. If at these large values of $\lambda$, introducing just a small effect of similarity produces a large fall in effective diversity from the maximal naive diversity, then the repertoire must contain many similar clones. Conversely, if such clustering produces only a small fall in diversity then the clones must be spaced further apart in the metric space. This measure is therefore a hybrid of monitoring the effect of clonal expansions and similarity simultaneously, scanning through the complexity of these two intertwined repertoire facets. Our measure is highlighted by the pink area in generated diversity profiles (Supplementary Note 2.2), defined as lying between the naive profile $\lambda = \infty$ and the profile for $\lambda = 16$. We have shown that, in the case of a uniformly distributed $CDR_\beta 3$s in a repertoire, with the increase of similarity between $CDR_\beta 3$s the area between the $\lambda$ curves increases (Fig. 3a. right panel and Supplementary Note 1.7).

In the case of natural repertoires the effect of clonal expansion will interplay with the similarity. The area between profiles for $\lambda = \infty$ and $\lambda = 16$, is presented for different treatment groups in Fig. 3b (middle panel). We concluded that there is a tendency to fall at the early time points after vaccination (5 and 14 days), with a further fall of comparable magnitude by day 60. As the area is influenced by $q$ it is also closely connected to the slope of diversity curves and therefore clonal expansion. Thus, the fall of value of area cannot be attributed to a decrease in similarity at later time points. When other repertoire features are taken into account, such as the slope and the trend of $\Delta ln(D(q,\lambda))$ at small $\lambda$, the decrease in area can be explained by the driving effect of clonal expansions at later time points. In conclusion, comparing the $\Delta ln(D(q,\lambda))$ at small and large values of $\lambda$ has enabled us to make some biological conclusions about the structure of the repertoires.

**Non-linear relationships between TCRDivER features are driven by repertoire structure.** As individual repertoire features clearly correlate to biological properties of TCR repertoires, we investigated relationships between individual features. There are many combinations of features extracted from diversity profiles that can be derived from similarity scaled diversity profiles, but for brevity we highlight in our opinion the most important ones (Fig. 3c). Firstly, while many of the feature correlation plots are clearly non-linear, they lie in a surprisingly tight area. This indicates that there is some constraint at work in the structure of the repertoires meaning that the value of one feature tightly constrains the value of other features. However, the second interesting characteristic is that these constraints are not unique.

An example is the relationship between $\Delta ln(D(q,\lambda))$ at small $\lambda$ and the area between profiles for $\lambda = \infty$ and $\lambda = 16$, as shown in the first panel of Fig. 3c. As noted above, these both depend on the distances between sequences in the repertoire. Rising $\Delta ln(D(q,\lambda))$ at small $\lambda$ indicates mean distances between sequences in the repertoire increasing. Smaller area between diversity profile curves at higher $\lambda$s indicates a decreased weighted $CDR_\beta 3$ sequence similarity within the repertoire. In a simple ideal case where the clone size and distance distributions are uniform

these two effects perfectly coincide. We illustrate this using model data in Supplementary Fig. 2. In real data with non-uniform clone size and distance distributions $D(q,\lambda)$ provides a measure where the differential effect of changes in the most similar sequences and changes in comparisons across the repertoire as a whole can be characterised. Upon closer examination, we can notice that there is a multidimensional separation to three groups in the feature correlation plot going from late timepoints with both features at low values, increasing through intermediate timepoints with both CFA and CFA+OVA treatments, finally stabilising at untreated values with both feature values relatively high indicating low similarity and clonality.

Another example is the relationship between the slope between $q = 1, 2$ for $\lambda = $ identity and the area between the curves at large $\lambda$ values for naive diveristy. Low values of slope, as discussed previously, indicate low clonal expansions. And since here $\lambda = $ identity there is no similarity taken into account. The other feature is dependent also on similarity as well as the clonal expansion. These two features combined give a clean separation of untreated vs treated samples. But interestingly, show also that many of the CFA+OVA treated repertoires also share lower clonal expansions with untreated samples as indicated by the slope (along the y-axis). Traditionally, additional antigenic exposure is correlated with higher diversity and stronger repertoire response. However, we might hypothesise that these repertoires might not show high clonal expansions as they are responding with a multitude of smaller and not so similar clones. And indeed, when looking at the placement of these repertoires along the correlation curve between $\Delta ln(D(q,\lambda))$ at small $\lambda$ and the area between profiles for $\lambda = \infty$ and $\lambda = 16$, as shown in the first panel of Fig. 3c.

This again emphasises the importance of taking multiple features of the diversity profile to more fully characterise repertoire structure. Similar non-unique constraints can be seen in the other panels of Fig. 3c.

The mathematical form of the diversity we have adopted does impose some restrictions on possible diversity profiles. For example, the effective diversity must always fall (or stay constant) as $q$ rises, reflecting the down weighting of small clones. To test if these relationships might be an artefact imposed by the mathematical form of the diversity we replaced the clone size distribution with pseudo-random numbers while keeping the distance matrix fixed (Supplementary Fig. 19). This eliminated previously observed correlations, showing that the correlations are not mathematically necessary. To confirm that the correlations are not a product of the particular distance definition adopted we repeated the analysis using an alternative metric based on amino acid properties using the biochemical scoring based on the Atchley factors[44]. This shows qualitatively similar correlations (See Supplementary Fig. 16).

To validate some of our findings we analysed a human TCR repertoire dataset previously published in[43]. In short, T cells were isolated from blood samples taken from non-small-cell lung cancer patients prior and post-treatment with CTLA-4 blockade (ipilimumab) in combination with radiation therapy (RT). The obtained T cells were sequenced in bulk. We analysed these repertoires as with the murine data, using TCRDivER to construct diversity profiles for analysis.

We found that pairwise relationships between features were quantitatively quite different that the murine data, but displayed the same characteristics of lying on curves and giving rise to non-unique constraints (Supplementary Fig. 29) Given differences in species, tissue and treatment, it is unsurprising that the range of repertoire structures observed differs considerably. At least some of these differences are captured in features of the similarity scaled diversity profiles. Despite these differences, the human data corroborates the notion that regardless of the immunisation

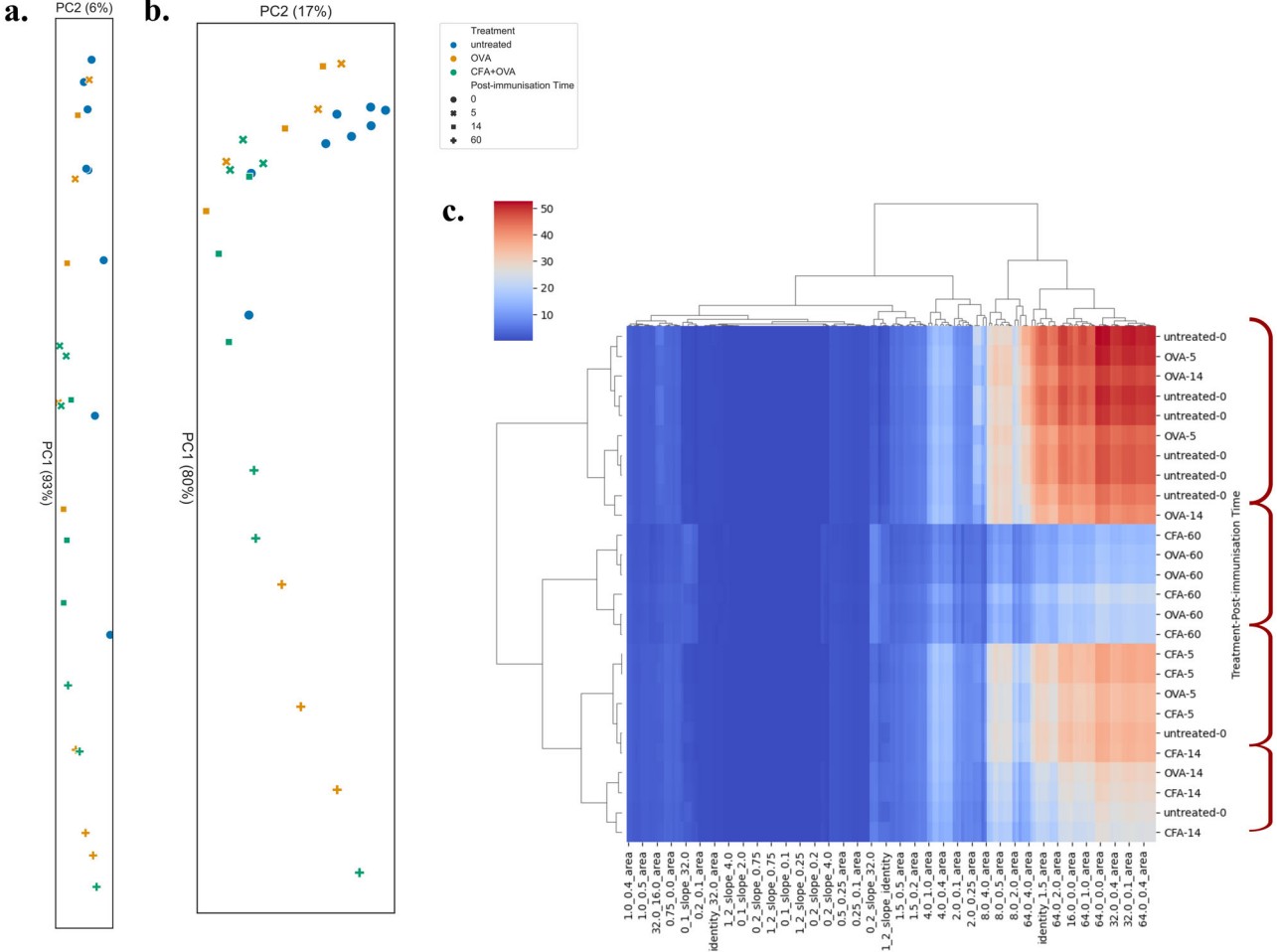

**Fig. 4 Clustering based on diversity estimates of samples from the murine dataset. a** PCA on naive diversity values $D(q)$, i.e., $\lambda = $ identity. The PCA plot aspect ratio has been adjusted and corresponds to variation explained by the first two principal components. **b** PCA on features extracted from the diversity profiles constructed on the diversity values $D(q, \lambda)$. These include areas between all lambda curves, average $\Delta \ln D(q, \lambda)$ for small $\lambda$s and slopes $q = 0 \to 1$, $q = 0 \to 2$ and $q = 1 \to 2$. As in **a**, the aspect ratio corresponds to variation found by PCA. The legend is shared between figures **a** and **b**. **c** Complete linkage hierarchical clustering of diversity profile features using Euclidean distances between the features. Clusters of interest are highlighted in dark red on the right side. A subset of features that have been used for the clustering are shown at the bottom.

strategy and dataset (human or murine), the natural TCR repertoires reside in a subspace governed by a complex interplay of TCR clonality and similarity.

**TCRDivER features improve separation of biologically distinct repertoires**. In order to test if the similarity-scaled diversity profiles can be used to classify repertoires we used principal component analysis (PCA) and hierarchical clustering. We carried out this analysis first on the values for the naive diversity profiles alone and then for the all values in the complete similarity-scaled diversity profile (Fig. 4a and Supplementary Fig. 10).

Both the naive and similarity scaled profiles show a strong PC1 which is driven by the expansion of large clones at the late day 60 time point. However, relative to the naive profile, PCA on the similarity scaled profiles shows more than twice the variance in PC2 i.e. more information along the second dimension of separation. It can be seen that this allows for substantial separation of immunised vs. untreated controls in the second dimension (Fig. 4c). If we imagine that TCR repertoires reside in a multidimensional space characterised by clone size distributions and similarity, then naive profiles allow for separation in the

clonality dimension, while similarity scaled diversity profiles allow for separation along both dimensions.

To test if the features we have identified are useful in capturing key dimensions of variation in the similarity scaled diversity profiles we then extracted these features and carried out PCA on the features rather the raw values. These include all the areas between different $\lambda$ curves, average values of $\Delta \ln(D(q, \lambda))$ for small $\lambda$s and slopes for $q = 0 \to 1$, $q = 1 \to 2$ and $q = 0 \to 2$. To further mitigate the effect of repertoire size on the analysis we have log-transformed the values of $D(q, \lambda)$ prior to feature extraction. The analysed features are therefore ratios and relationships of the natural logarithm of $D(q, \lambda)$. In both the mouse and human datasets we found that the PCA on features was qualitatively similar to that on the full diversity profile (Fig. 4b and Supplementary Fig. 28). However, the variance explained by PC1 was reduced while that explained by PC2 was increased, leading to improved 2 dimensional separations. In the case of the mouse data the effect was modest, while in the human data it was more substantial, leading to an increase in variance explained by PC2 from 11% to 20%. The increase of variance explained in PC2 indicates that these features are indeed acting as useful summaries of redundant (linearly correlated) information in the full profile. Therefore making use of such features, rather

than the raw profile values may help reduce experimental noise and improve robustness.

Similarly, hierarchical clustering of diversity features shows strong separation between untreated samples, as well as early and late time point repertoires of the murine dataset (Fig. 4c). This showcases the robustness of using diversity features within different stratification techniques. Furthermore, it strengthens the idea that these features can be subjected to more sophisticated analysis such as machine learning approaches.

In both the datasets analysed (human and murine) the number of samples per category was low. Nonetheless, we have employed a leave one out machine learning approach to assess the accuracy of predicting immune status using similarity scaled diversity profiles. Previously, Greiff et al.[39] show that by inputing naive diversity profiles into a linear Support Vector Machine (SVM) classifier they were able to achieve a predicted accuracy in the range of 80–90%. However, we would like to note that the dataset in the article by Greiff et al.[39] contained repertoires from 24 patients samples at 3 timepoints, thus enabling for a proper split into train and testing sets and enabling the model to correctly learn the signal. The authors of this article find that the data from Greiff et al.[39] is no longer publicly available. We have therefore employed an SVM classifier and a Random forest classifier on the previously analysed murine samples, as this dataset is the main focus of this article.

Due to low sample size per group we have attempted classification on two parameters: timepoint (day 0, 5, 14, and 60) and treatment (Untreated, CFA, CFA + OVA). We have input similarity scaled diversity and naive diversity profiles retaining the same dataset partitioning (see section Classification using SVM and random forest. Using the SVM classifier we were able to obtain 0.81 average predicted accuracy on similarity scaled compared to 0.85 on naive diversity profiles. We believe that this is due to the number of large clones present at later timepoints. When using similarity scaled diversity the algorithm performs worse, possibly due to diversities calculated at smaller values of lambda. When the same classifier is employed on treatment, the predicted accuracy is higher for the similarity scaled (0.73) versus naive (0.65) diversity profiles. We believe that this flip in performance is due to the algorithm capturing signal on similarity which is necessary for prediction solely on treatment, as each treatment category now contains all timepoints. When employing Random Forest, the predicted accuracy is higher in similarity scaled (0.77 for both timepoint and treatment classification) versus naive diversity profiles (0.69 for timepoint and 0.54 for treatment).

We attribute the overall lower accuracy of prediction compared to what Greiff et al.[39] report to two aspects: one is the previously discussed sample size, and the other that there was no feature selection step prior to employing classifiers. Therefore non-informative diversity indices were also input into the classifiers which can reduce performance. We overall, find the performance of both classifiers for treatment prediction to be promising. We do not encourage the use of classifying algorithms on such small dataset as the ones analysed in this article. However, we do show that even with obstacles, similarity-scaled diversity profiles give an additional level of information that allowed a higher classifying performance compared to naive diversity in both algorithms employed.

## Discussion

The complex structure of immune repertoires makes them challenging to compare and classify. Previous work making use of sequence information to understand TCR repertoires has focused on determining the antigen specificity of particular sequences. In contrast, TCRDivER is able to make use of sequence information to reveal structural similarity between repertoires that have little or no sequence overlap. With two tunable parameters TCRDivER becomes an effective computational microscope, able to focus on different scales of structure in the immune repertoire. The resulting diversity profiles then provide highly interpretable summaries of global multiscale structure.

Here we provided a proof-of-concept study in application of similarity scaled diversity estimates to TCR repertoire analysis. We have applied principal components analysis and hierarchical clustering as a means of qualitative repertoire stratification and shown it is able to capture variation in the diversity profiles which characterise immunisation history. The similarity scaled diversity profiles are themselves quite rich summaries and in the future we anticipate that they may be subject to more sophisticated machine-learning techniques. At present this possibility is limited by the number of available samples for which comparable data is available.

As has previously been shown by Grieff et al.[39] the use of a single diversity index to rank or classify repertoires is not robust, since the ranking will depend on the index selected. The use of naive diversity profiles is a step forward in so far as they reflect the contribution of both large and small clones. However, when applied to real TCR data naive diversity profiles typically give a single effective dimension of separation. This is reflected in our results that show 93% of variance explained by PC1 carried out on the naive diversity profile. By taking into account sequence similarity simultaneously with clone size distribution our similarity-scaled diversity profiles provide a genuine second dimension of variation in the structure of the repertoires. As shown in our PCA analysis this opens the practical possibility of 2D separations of repertoires which are inherently more powerful. Again, this approach can make use of sequence data to classify repertoires together even when they share no similar sequences. This attribute of the algorithm can allow to investigate "public" TCR responses to disease-related antigens not in terms of individual shared TCRs, but in terms of aforementioned "public repertoire structures".

The striking relationships between similarity profile features appear to reflect biological structure in the TCR repertoires. We hypothesise that the non-linear relationships we observe reflect the range of possible biological variation and are thus analogous to the way in which clone size distributions are well approximated by power laws. Our observation motivates investigations of extending power law distributions to include description of sequence similarity.

Since the algorithm still relies on an all-against-all comparison, there is scope for several-fold speed up, including optimisation of the distance function. A possible gain would come from replacement of the exact all-against-all comparison at high lambda with comparison against approximate k-nearest neighbours using a ball tree algorithm. However, at low lambda values the all-against-all comparison cannot be avoided reflecting the way in which the similarity-scaled diversity incorporates genuinely global information about the repertoire.

The form of equation in Fig. 1b. is motivated by rather general mathematical considerations, but these still leave the metric used to compare sequences undetermined. There is no 'true' metric, in the sense that a assigning a single number to the distance between two sequences cannot fully capture all the ways in which binding affinities vary. In our work we used two metrics which plausibly reflect biological functional similarity (through use of evolutionary data in BLOSUM45 matrix) and biochemical similarity (through the Atchley factors). While these gave qualitatively similar results, the best metric to use for a given question is an open question in the field of TCR analysis as a whole.

Here, we would also like to emphasise that sequencing depth also influences conventional diversity estimates as absolute, not relative, clone frequencies are needed for diversity estimations[3]. Increasing sequencing depth maximises the number of reads per T cell clone and therefore allows for capturing of a more complete TCR repertoire[3,45,46]. Experimentally, repertoires from different laboratories or experiments will often be sequenced at a different sequencing depth which makes them difficult to compare using conventional diversity estimates. TCRDivER subsumes most conventional diversity estimates and instead of comparing absolute diversity numbers between repertoires, it focuses on the relationships between these estimates within the repertoire. Therefore, it in some ways bypasses the need for absolute quantitation of TCR frequencies in the repertoire. We are aware that influence of the sequencing depth and experimental sampling still persists in the raw repertoires as the number of unique clones varies between repertoires in both datasets (Supplementary Tables 1 and 2). Thus we have chosen to subsample the $CDR_\beta3$s of the repertoires based on the original frequency distribution and mitigate the influence on the absolute numbers of unique clones (richness) on the diversity estimates. This also normalises the computational cost of each repertoire calculation to under 6 hours, which we find important for practical applications of TCRDivER.

While in this study we have applied similarity-scaled diversity profiles to TCR repertoires, we believe that the same concept should also be applicable for understanding antibody repertoires. The features of similarity-scaled diversity profiles can easily be translated in to properties of the repertoire. The functional biological importance of the similarity scaled diversity (as indeed the naive diversity) is likely to be more variable and subject to experimental investigation. TCRDivER provides an important tool to enable those investigations.

## Methods

**Data acquisition and description**. The murine dataset consits of previously published data that has been analysed as part of a larger dataset in the work by[21]. Briefly, CD4[+] T cells were isolated from spleens of 18 C57BL/6 mice immunised with Complete Freund's adjuvant (CFA) with or without an addition of Ovalbumin antigen (OVA). The samples were collected at different times post-immunisation: at day 5 and 14 (early timepoints) and day 60 (late timepoint). In addition, CD4[+] T cells were collected from 8 healthy unimmunised mice prior to study start. An overview of the dataset is given in Table 1. We have received the dataset already analysed with Decombinator, which described in depth in[47,48]. The data we have analysed consisted of a list of $CDR_\beta3$ sequences present in each sample. The raw fastq files are available at http://www.ncbi.nlm.nih.gov/sra/?term=SRP075893. A full data overview as well as any supporting results concerning the murine dataset are provided in Supplementary Note 2.

To validate the observation of improved PCA separation, we analysed an additional dataset of human TCR repertoires. Additionally, we have analysed a human TCR dataset previously analysed by[43]. The participants of the study were 39 patients diagnosed with metastatic non-small-cell lung cancer (NSCLC). They were treated with daily radiation therapy regimen in two phases of the trial

(phase I - 6Gy × 5 and phase II 9.5 Gy ×) and intravenous ipilimumab (CTLA-4 blockade) following the first radiation treatment and subsequently repeated every 3 weeks for four cycles. The assessment of patient treatment response was performed with PET/CT scans at day 88 and evaluated using Response Criteria In Solid Tumors (RECIST). The patients were then classified, according to RECIST, as complete responders (CR), partial responders (PR) with tumour decrease in size ⩽ 30%, stable disease (SD) with insufficient shrinkage to qualify for any of the other criteria, and progressive disease (PD) with increase in size > 20% or appearance of new lesions. Out of 39 patients 20 were evaluable at day 88. Serial blood samples for peripheral blood mononuclear cells (PBMCs) were collected at baseline (day 0), and on days 22, 43, 64, and 88. The isolated PBMC were subjected to amplification and sequencing of bulk $TCR\beta$ $CDR_\beta3$ regions by Adaptive Biotechnologies. We have obtained the data from the Adaptive Bioctechnologies ImmunoSEQ database[49]. Since the samples collected at later timepoints (day 43 and onward) were not available for all of the 20 evaluable patients, we have restricted our analysis to samples collected at baseline and day 22 of treatment. An overview of samples included in our analysis is given in Table 2. We analysed the data as before by calculating the diversity $D(q,\lambda)$ and constructing diversity profiles (Supplementary Note 3.1). All of the data overview and supporting results of the analysis of the human dataset are given in Supplementary Note 3.

**Subsampling to reduce computational load**. We have only considered "In" frame reads of $CDR_\beta3$s in our analysis. In order to reduce the computational load of calculating pairwise similarity between a large number of $CDR_\beta3$ regions (order of magnitude ≈ $10^5$), we have performed subsampling prior to analysis. We will refer to individual $CDR_\beta3$ sequences as sequences, and a collection of identical $CDR_\beta3$ sequences as a clone. Each repertoire was down sampled to contain 50000 $CDR_\beta3$ sequences. The sampling was random, based on the original $CDR_\beta3$ frequency distribution. After sampling the number of $CDR_\beta3$ clones was ⩽ 50,000. An overview of number of unique clones for the murine and human dataset is given in Supplementary Tables 1 and 2, respectively. For each sample, the $CDR_\beta3$ clone count (number of identical $CDR_\beta3$s within a clone) was transformed into frequencies, so that the all the $CDR_\beta3$ clone frequencies within a repertoire sum up to 1:

$$\sum_{i=1}^{S} p_i = 1, \tag{1}$$

where $p_i$ is the frequency of the $i$-eth $CDR_\beta3$ clone in the repertoire. The clone $CDR_\beta3$ amino acid sequences with their respective frequencies were used as input further downstream.

**Calculating the distance matrix**. In order to reduce computational time and memory usage of calculating pairwise comparison, we have divided the distance matrix into smaller portions ("chunks") which are separately calculated. The algorithm takes in the list of $S$ $CDR_\beta3$ clones, and splits it up in to $n$ sub lists of predefined length, default value is 100. For each sublist pairwise comparisons are made between the $CDR_\beta3$s in the sublist versus all the $CDR_\beta3$s in the original list (Supplementary Fig. 1). Global alignment between two $CDR_\beta3$ clone sequences was performed with a gap penalty of 10 and scored using the BLOSUM45[50] substitution matrix. The alignments were created using the PairwiseAligner function in the Bio.Align package within Python3[51]. The distance between two $CDR_\beta3$s, $d(CDR_\beta3_i, CDR_\beta3_j)$, was calculated based on the alignment scores:

$$d(CDR_\beta3_i, CDR_\beta3_j) = 1 - \frac{BLOSUM45score(CDR_\beta3_i, CDR_\beta3_j)}{max(BLOSUM45score(CDR_\beta3_i, CDR_\beta3_i), BLOSUM45score(CDR_\beta3_j, CDR_\beta3_j))} \tag{2}$$

Alternatively, we have also employed a biochemical scoring based on the Atchley factors[44]. The factors are based on biochemical properties of amino acid

---

**Table 1 Murine Dataset overview: Overview of samples available in the analysed dataset, a part of a previously published dataset by[21].**

| Treatment | Sample collection time (days) | Number of mice |
|---|---|---|
| CFA | 5 | 3 |
| CFA | 14 | 3 |
| CFA | 60 | 3 |
| CFA+OVA | 5 | 3 |
| CFA+OVA | 14 | 3 |
| CFA+OVA | 60 | 3 |
| Non-immunised | 0 | 8 |

In total 26 CD4[+] murine spleen samples were analysed. Sample collection time is given as the number of days post immunisation. Note that mice culled at day 60 received an additional booster shot of immunising agent (CFA or CFA+OVA).

---

**Table 2 Human Dataset overview: Overview of samples used from the dataset previously published by[43].**

| RECIST criteria | Sample collection time (days) | Number of patients |
|---|---|---|
| PD | 0 | 8 |
| PD | 22 | 8 |
| SD | 0 | 5 |
| SD | 22 | 5 |
| PR | 0 | 5 |
| PR | 22 | 5 |
| CR | 0 | 2 |
| CR | 22 | 2 |

In total 40 human PMBC samples were sequenced for $TCR\beta$ CDR3s. Sample collection time has been given as the number of days after start of therapy, with 0 being baseline prior to first treatment.

residues that are grouped and transformed into five Atchley factors. For each $CDR_\beta3$ an average value for individual Atchley factors was computed. The distance between two $CDR_\beta3$ sequences is then calculated as an Euclidean distance between the averaged five Atchley factors.

The obtained distance matrix is mathematically connected to the similarity kernel Z as shown in equation (3)

$$Z_{ij} = e^{-\lambda d_{ij}}, \quad (3)$$

where the $\lambda$ provides scaling, and $d_{ij}$ the distance between two $CDR_\beta3$ clones.

**Calculating naive diversity**. Naive diversity does not take similarity into account and is calculated based on the frequencies of $CDR_\beta3$ clones within the repertoire[34,38]:

$$D(q) \equiv \left( \sum_{i=1}^{S} p_i^q \right)^{\frac{1}{(1-q)}}, \quad (4)$$

where $p_i$ is the frequency of the $i$-eth $CDR_\beta3$ clone and $q$ is the diversity order, as diversity indices are a functions of $\sum_{i=1}^{S} p_i^q$. We have chosen a list of $q$s which subsume the use of some common diversity indices such as 0, 1, 2, and $\infty$, which correspond to the richness, exponent of the Shannon diversity[32], Simpson[33,34] and Berger-Parker[35] index, respectively. To extend our surveying the clone size distribution space we have also added 3, 4, 5, and 6th order of diversity. For values of $q = \{1, \infty\}$ the diversity was calculated as the limit of $q$ approaching the values of 0 and $\infty$.

$$^\infty D = \frac{1}{p_{max}} \quad (5)$$

$$^1 D = e^{(\ln{}^1 D)} = e^{(-\sum_{i=1}^{S} p_i \ln p_i)} \quad (6)$$

A detailed derivation of the equations is provided in Supplementary Notes 1.1, 1.2, 1.3, 1.4, and 1.5.

**Calculating similarity-scaled diversity**. Similarity scaled diversity of order $q$ takes $CDR_\beta3$ clone sequence distances $d(CDR_\beta3_i, CDR_\beta3_j)$ along with their respective frequencies. We have adapted the method of calculating similarity-sensitive diversity measures, as proposed by[41]. We have again chosen a list of $q$s, $q = 0, 1, 2, 3, 4, 5, 6, \infty$, in the same manner as when calculating the naive diversity. Furthering the method,[41] propose the use of similarity-sensitive diversity measures $D(q, \lambda)$ (Equation (7)), dependent on relative abundances and species similarity data as distance $d_{ij}$. We introduced an alteration of the original approach is introducing the similarity scaling factor $\lambda$, which allows us to weight TCR distances in much the same way as we do clone sizes (Fig. 1d). Here we choose a list of values $\lambda = \{0.0, 0.1, 0.2, 0.25, 0.3, 0.4, 0.5, 0.75, 1.0, 1.5, 2.0, 4.0, 8.0, 16.0, 32.0, 64.0\}$. The value of $\lambda = $ identity corresponds to the naive diversity calculation, $D(q)$.

$$D(q, \lambda) = \left( \sum_{i=1}^{S} p_i (\mathbf{Zp})_i^{q-1} \right)^{\frac{1}{(1-q)}} \quad (\mathbf{Zp})_i = \sum_{j=1}^{S} Z_{ij} p_j \quad Z_{ij} = e^{-\lambda d_{ij}} \quad (7)$$

**Downstream analysis of TCRDivER output**. The final output of TCRDivER is a table containing all the values of diversity calculated with a range of $q$s and $\lambda$s. The full downstream analysis is summarised in a Python3.6 jupyter notebook[52] available alongside the main TCRDivER algorithm at https://github.com/sciencisto/TCRDivER. The diversity profiles were constructed using the seaborn package[53]. Areas between $\lambda$ curves of $\ln(D(q, \lambda))$ was calculated within the numpy framework[54] as an integration using the composite trapezoidal rule. Average $\Delta \ln(D(q, \lambda))$ for small $\lambda$s was calculated for $\lambda = \{0.0, 0.1, 0.2, 0.3, 0.4, 0.5\}$ as the mean of the average difference between $\ln(D(q, \lambda))$ at each calculated $q$. The range of $\lambda$s in this analysis is chosen for higher and lower values separately. For the higher values, the first lambda is chosen as 1, which equals to no similarity scaling. The value of $\lambda$ is then doubled until $\lambda = 64$, as in a geometric sequence. For lower values $\lambda < 1$ has been incremented by 0.1 for values of 0.1 to 0.5 and 0.25 and 0.75 values were added as well. The users of this algorithm are free to choose their own values by augmenting the configuration file. Given the exploration of mathematics of similarity scaled diversity estimates, we feel that the range captures repertoire information sufficiently. Due to time constraints we have not yet devised an algorithm for choosing the minimum number of *lambda* values needed.

Slopes of diversity when $q = 0 \rightarrow 1$, $q = 1 \rightarrow 2$ and $q = 0 \rightarrow 2$ were calculated as differences between the diversities when $q = 0, 1$, and 2. We are aware that this is not the slope of diversity at the specific values of $q$, as we also provide the mathematical evaluation of the slope at these timepoints (Supplementary Note 1.1 and 1.2). However, due to time and memory considerations we have opted for the simplified calculation as we feel that it represents the slope sufficiently. For future implementations, we will update the algorithm to include the analytical slope evaluation. Principal components analysis was performed as implemented in the scikit-learn package[55].

**Table 3 Murine Dataset Partitions.**

| Treatment | Sample collection time (days) | Total number of samples | Number of samples per partition | | |
|---|---|---|---|---|---|
| | | | I | II | III |
| CFA | 5 | 3 | 1 | 1 | 1 |
| CFA | 14 | 3 | 1 | 1 | 1 |
| CFA | 60 | 3 | 1 | 1 | 1 |
| CFA+OVA | 5 | 3 | 1 | 1 | 1 |
| CFA+OVA | 14 | 3 | 1 | 1 | 1 |
| CFA+OVA | 60 | 3 | 1 | 1 | 1 |
| Non-immunised | 0 | 8 | 3 | 3 | 2 |

**Classification using SVM and random forest**. Both Support Vector Machine (SVM) and Random Forest classifiers were employed on the similarity scaled and naive diversity of repertoires on the murine dataset. The classifiers were used as implemented in the scikit-learn[56] package available for python. The datasets contains samples from CFA and CFA+OVA treated mice at day 5, 14, and 60, as well as untreated samples from day 0. With the exeption of the untreated samples ($n = 8$) there only 3 samples per each timepoint-treatment combination. Therefore to increase the number of samples classification was attempted based on two categories: treatment (untreated, CFA, CFA + OVA) and timepoint (days 0, 5, 14, and 60). Due to the size of the dataset, there was no part of the data left as a validation set. Instead the dataset was divided into three partitions to contain a sample from each timepoint-treatment group, as shown in Table 3. The classifiers were employed on the partitions in a leave one out manner, so that one partition was used as a test set and the remaining two as training sets. All of three predictions were then concatenated (each partition was used once as a test) and an average accuracy was calculated. The code is available in the downstream analysis jupyter notebook along with the main TCRDivER algorithm https://github.com/sciencisto/TCRDivER.

**Statistics and reproducibility**. In total, 26 CD4$^+$ murine spleen samples were analysed in the murine dataset[21]. The samples are separated by time of collection and treatment. There are 3 samples per each combination of treatment and time, with the exception of 8 samples for non-immunised mice. In the human dataset[43] a total of 40 samples have been analysed. The samples are separated by collection time and RECIST criteria. An overview of samples is shown in 2. All analysis in this article can be reproduced either from raw data supplied in the github repository or original sources as stated in the Data availability section. Downstream analysis of TCRDivER outputs needed to reproduce the figures and analysis in this article are available via jupyter notebooks within the TCRDivER github repository https://github.com/sciencisto/TCRDivER.

**Reporting summary**. Further information on research design is available in the Nature Portfolio Reporting Summary linked to this article.

## Data availability

Both datasets analysed in this article come from previously published studies. The murine dataset[21] raw fastq files are available at http://www.ncbi.nlm.nih.gov/sra/?term=SRP075893. The human dataset[43] is available on the Adaptive Biotechnologies ImmunoSEQ database[49]. Furthermore, all data needed to reproduce the figures and analyses in this article is available in the TCRDivER github repository https://github.com/sciencisto/TCRDivER.

## Code availability

The code is available at https://github.com/sciencisto/TCRDivER.

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

## Acknowledgements

We would like to thank Prof. Nir Friedman, Dr. Shlomit Reich-Zeliger, and Dr. Erik Shifrut from the Weizmann Institute, Rehovot, Israel for generating and sharing the TCR sequence data analysed in this paper.

## Author contributions

M.V. designed and developed TCRDivER algorithm, wrote the code, ran theoretical experiments, designed the figures, wrote the original draft. J.K. designed and developed TCRDivER algorithm. J.K., P.M., and T.L.A. supervised the research. T.L.A. acquired research funding. B.C. contributed with data acquisition and research feedback. All authors contributed to writing the manuscript.

## Competing interests

The authors declare no competing interests.
