## [Peer Review File · Communications Biology]

Reviewers' comments:

Reviewer #1 (Remarks to the Author):

In this interesting paper, the authors describe an algorithm that analyzes in depth the diversity of TCR repertoires, combining analysis of sequence similarity and global diversity of the repertoire. Both aspects are related to function of the immune repertoire and together constitute a good approach to characterizing its dynamics in response to pathogenic invasion or cancer development. I would like in particular to salute the effort the authors made to intuitively explain complex relationships between features of the similarity scaled diversity profiles.

This algorithm could be an important step in overcoming a major difficulty of repertoire comparison between individuals: the fact that individuals develop different clones to fight the same threats.

The authors describe in depth the algorithm, its application to two datasets and derive biological insights about repertoire dynamics using mathematical tools. In summary, the main novelty of the paper is three-fold:

- developing a way to combine sequence similarity and diversity into a two-parameter function
- developing the algorithm that implements this function and summarizes the results
- defining features of the algorithm output that have biological relevance

Overall, the approach is new and promising and the author manage to explain it fairly intuitively. However, I find that where they fall a bit short is in proving quantitatively its superiority for predictive/classification purpose. They do little to compare the output of the algorithm to other methods.

Their claim is that the data is insufficient to reach conclusions about the predictive power of the algorithm. I write below specific comments and requests for clarifications, I recommend publication should the author be able to address them:

1. In the summary and introduction, several references are made to repertoire changes, implying changes through time. However, I find some of these references a bit confusing because the algorithm is made to look at snapshots of the repertoire. Sure, it is built to compare statistics of repertoires and as such it can help compare repertoires from the same individual at different time points, but no more than from the different individuals, as far as I understand. I think it would be important to somehow make it clear that at no point the algorithm puts together data from different time points.
2. In the introduction in the sentence "T cells activate and proliferate upon epitope-specific contact, creating a complex mix of receptors aimed at the antigen of origin." I find the end of the sentence very confusing and I am not sure what you mean.
3. In general, I find that the authors tend to mix static description of the repertoire (some clones are large, others are small) with dynamical descriptions, in particular expanded clones. While it is true that a large clone has at some point expanded, because in part of the paper the authors make comparisons between different time points I find it confusing to use the term expanded to describe a clone that is simply large. This sort of goes together with point 1 as a need to make a clearer distinction between analysis of a snapshot, and analysis that leads to interpretation in terms of dynamics.
4. Here I come to the central issue I have with the paper in its current state, being that there is a lot of qualitative justification for the algorithm, but no quantitative comparison with other algorithms or other options. One first question: for a given value of λ , varying q ends up being a whole function or the real variable. Even if you only compute it for a finite number of values of q it is dimensionally as large as computing the clone size distribution, which arguably contains as much information as the transformed diversity estimate. So, if one were to cluster, for each value of λ , clones together in a smart way (and here I have to admit I don't have a definite answer to what this smart would be) and then use instead of the q -based diversity measure the clone size distribution for that value of λ , it would be much faster to compute. Would the output be less interpretable and predictive? Could such a comparison be made? Is the only issue that it is difficult to cluster clones based on similarity or is there is another why not to go down that road?
5. I think the idea that sequence similarity leads to closer function should be discussed in a bit

more depth, in particular a discussion of the literature and of any quantitative arguments here would be welcome.

6. "Individual repertoire analysis begins with CDR 3 sequence ltering and subsampling in order to reduce computational cost and

keep only sequences of interest e.g. such as those that generate functional amino acid sequences." if anything else is removed here, it would be important to mention it, if not it can also be stated clearly.

7. When mentioning the "chunks" division in the main text, could you clarify if those are picked at random or structured?

8. Issues regarding sequencing depth are mentioned quite often, and it also written that the diversity profiles you pick overcome some of these issues. I think it would be worth a bit of discussion (describing the issues and how they are overcome) and, if it is already in the literature then just a summary of intuitive reason why similarity scaled diversity helps with these issues.

9. When you describe picking 50000 sequences at random in the main text and in figure 2 I found it a bit confusing and was not able to understand if you were picking 50000 sequences or 50000 unique sequences. However, I found the explanation in the methods perfectly clear. Could you try and clarify in the main text and caption of fig2?

10. Could you discuss a bit the scale of lambda? I understand that you argue for a range of lambdas, but when picking values to evaluate, how do you decide? What are values of lambda that can be argued to matter in terms of immune function? For instance why do you pick 16 as the highest point before infinity? Maybe your algorithm is a way to find out? I think this discussion would be very interesting for all readers, from clinicians to computer scientists.

11. "We observed that for small values of lambda the curves are approximately flat and are evenly spaced on a log scale in all diversity profiles" -> link to figure where it is shown would help here

12. Again in the interest of comparison, for repertoire change following immunization, how does your method compare with looking at for instance fold change? There have algorithms developed for that. I understand that the advantage of your method is that you can compare repertoires both longitudinally and between individuals, but it would be important to state for what applications that is essential.

13. In figure 3B I find the choice of x axis ticks very strange. Why put in a line things that are both ordered in time and ordered by antigen? Why not have several lines then? Or different colors? At the very least not a line plot where there is no clear ordered progression along the x axis.

14 In figure 3C what do the colors represent?

15. Regarding hierarchical clustering, you claim that there is very strong separation between different samples. While it is clear that the clusters are much better than random, it is difficult to assess the quality of the clustering without a quantitative point of comparison. Is there any way to think of one?

16. Regarding the PCA, you mainly talk about the variance explained by the dimensions, or refer to visual arguments about separation. Could you make the ability of the PCA to separate data quantitative? Is there a way to compare it to a more naive version quantitatively in its ability to predict outcome for human data or separate timepoints or treatment for the mice? For example, I find the analysis of relationship between features very convincing and quite complete. Is there enough data points to try some regression? decision tree? any classification? what about supervised PCA?

17. Link to previous point but you claim that the issue is the lack of data, if so, can you quantify how many repertoires you would need to have a reliable test of predictive power on PD/SD CR/PR groups for instance? In any case I think the claim that data is insufficient to test out the algorithm needs to be discussed a bit more. What about for instance the dataset in ref 38?

18. Regarding the choice to reduce the number of sequences to 50 000. Can you explain why that number? The complexity of the algorithm, if I understand correctly depends on the number of clones, not the number of sequences. So, you have to have picked that number based on estimated number of clones. Was it picked for a worst case scenario (i.e clones of size one)? Or based on average number of clones for 50 000 sequences?

19. In section 5.3 you mention that you use these chunks to reduce computational complexity and memory usage. Memory usage seems clear to me here, but I am not sure of how using these chunks reduces complexity. Could you specify? Don't you end up having to compute the whole matrix anyway?

20. The paper builds heavily on existing ideas regarding diversity measures, and the papers are generally properly cited. One missing point is that in ref 38 there is already mention of using

L'Hôpital's rule to find in particular the link to Shannon's entropy. I think to mention it in the SI.

21. In the SI, given that some calculations are a bit long, it would be good to make it self-standing and define all functions. I think that Z in particular is not defined (unless I missed it). Numbers for some equations would also make it easier to read in my opinion.

22. In the last equation of part 1.3, $q=1$, I think it would be clearer to write it.

23. In section 1.6, last equation of the series of three, I think the i index near the parenthesis is not necessary given that you are summing already and index is defined by summation right? If I am correct then keeping it makes it confusing.

24. You wrote that $\ln(1-x)$ is equivalent to x which is not correct.

25. "Note that the final form of the evaluation of $D(q; \lambda)$ for $\lambda \rightarrow 0$ is independent of the order of diversity q . It is solely dependent on the distance between CDR3 sequences weighted by their respective frequencies." Does it remain correct when q approaches 0 or infinity?

26. For the tables showing the profiles in the SI (example is table 14), is there a way to put some of the graphs from different mice, patients or time points (not all) next to each other so that they can be more easily compared? Otherwise, since the reader is not used to this type of graph, I find a bit difficult to compare.

Finally there are a few typos in the text which, once cleared will make reading smoother. A few here: give is written twice in intro, 7 weirdly printed in caption of fig 2, reperotires instead of repertoires a couple of times, in to/into, "is" missing in 5.1, more important some parenthesis typos in the SI that can be misleading

Reviewer #2 (Remarks to the Author):

Vujović et al. presented a newly developed algorithm, TCRDivER, as a method for TCR repertoire comparison based on diversity profiles, clone size and sequence similarity. Despite interesting and potential applications, there are some concerns.

Major issues:

Authors decided to use human TCR sequencing data from cancer patients to validate their findings in immunized - or not - mice repertoires. I understand that the aim was to test the new algorithm rather than focusing on sequencing data, however authors may explain why they used human sequencing data from cancer instead of repertoires from patients affected with autoimmune disorders or generally immune-mediated diseases, that would have matched with immunized murine repertoires more logically. Comments on how this choice may have impacted data analysis could be added in Discussion.

TCRDivER documentation is not clearly written and hard to follow for a non-computational person. Ideally, 2 pipeline examples should be shown — one using python scripts (local version) and one using bash script (server version). Also, the authors should provide the exact format of input data (what is the column order in tsv file? Does the column names matter?) and a test dataset.

All bash scripts contain a local path `"/work3/milvu/"` and thus can not be easily run on another server. The github repository contains an explanation how to fix this issue, but it is unclear (see section "specify working directory path - same as folder that is created in the filtering step (I.) on github <https://github.com/sciencisto/TCRDivER>"). All bash scripts should take a working directory as a parameter.

TCRDivER consists of multiple python or bash scripts (sometimes even a bash loop, see "II. Run all distance matrix calculations on <https://github.com/sciencisto/TCRDivER>"). The authors should wrap these scripts together in one script or command line tool that takes different arguments as input and not change these arguments in each script directly.

On the first step of the TCRDivER pipeline, the authors subsample the number of receptors to

50000. The question is: (1) why does repertoire subsampling not dramatically impact the similarity scaled diversity and (2) why 50000 sequences is enough to represent the diversity? Is the TCRDivER method robust to subsampling, i.e. are two diversity profiles constructed based on two different subsamples of one repertoire similar? Is it correct to compare two diversity profiles constructed based on different number of receptors, e.g. one repertoire contains 50000 receptors and the other only 20000 receptors?

Figure 2: in the caption clone size or frequency is mentioned, however the figure does not show any frequencies on steps I and II. What is "d" on step III?

Page 9: "While many of the feature correlation plots are clearly non-linear, they lie on surprisingly tight curves" — any set of points lie on a curve connecting these points. The authors may formalize their statement and ideally perform some sort of numerical analysis to prove it. Also, it is not clear that in Supl11b and Supl11d and thus the validation statement for the murine data is questionable.

Section TCRDivER features improve separation of biologically distinct repertoires. The authors should also compare their method to the original diversity profile methods.

Minor issues

The analysis of TCR sequence similarity architecture, diversity and antigen specificity with publicly available TCR datasets has been performed in several recent studies about autoimmune diseases and murine repertoires that may be cited at least somewhere in the Introduction, e.g. 1) doi: 10.7554/eLife.22057 and 2) <https://doi.org/10.1016/j.ebiom.2021.103429>.

Page 2 "Entropy, will give give more weight" — typo

Page 9: "There are many combinations of features that can be derived from similarity scaled diversity profiles" — which features?

Page 10 "This hypothesis seems confirmed when looking at..." — the authors should perform a statistical test or rephrase this sentence.

Response to Reviewer 1:

We greatly appreciate the detailed comments from Reviewer 1 and especially the in-depth summary of our work. We appreciate the opportunity to further improve the work. We hope that we have answered the comments in a satisfactory way.

Comment 1. In the summary and introduction, several references are made to repertoire changes, implying changes through time. However, I find some of these references a bit confusing because the algorithm is made to look at snapshots of the repertoire. Sure, it is built to compare statistics of repertoires and as such it can help compare repertoires from the same individual at different time points, but no more than from the different individuals, as far as I understand. I think it would be important to somehow make it clear that at no point the algorithm puts together data from different time points.

Our response: We agree with the reviewer and have made the changes in the summary and introduction highlighted in yellow in the main text. The reviewer is correct that we do not include longitudinal data in our analysis, therefore we removed most text containing "tracking" or "changes", and have substituted with investigating clone sizes or similar in the main text. We have also added two sentences in the introduction aiming to clarify that the algorithm takes in snapshots of the state of the repertoire:

Page 2. *Providing the complexity of the immune response is encoded both in sequence similarity and TCR clone sizes, such an analysis could allow for applications in immune monitoring e.g. following immunisation, therapy at different points in time or disease progression by comparison of "snapshots" of the repertoire state.*

Page 3. *TCRDivER compares similarity scaled diversity of snapshots of repertoires stemming from different immunological settings, be it different treatments, diseases, timepoints or their combinations. We would like to note that even though the algorithm can be used to compare repertoires from different timepoints, it at no point takes in longitudinal data as input.*

Comment 2. In the introduction in the sentence "T cells activate and proliferate upon epitope-specific contact, creating a complex mix of receptors aimed at the antigen of origin." I find the end of the sentence very confusing and I am not sure what you mean.

Our response: We agree and the sentence has been changed to "T cell activate and proliferate upon epitope-specific contact, thereby increasing their clone size". The confusing end of the original sentence was aimed to capture that multiple TCRs can bind a single epitope. T cells responding to the single epitope can have different TCRs and could also proliferate to a different extent upon activation. However, the way it was stated was overly complicated therefore we have tried to break up these complexities of T cell responses throughout the introduction. The statement that a single epitope can be bound by multiple TCRs was present already in the original text and we have expanded it with an additional sentence:

Page 2. "Moreover, T cells are cross-reactive meaning that a single TCR can bind multiple epitope targets. However, T cells responding to the same antigen might not proliferate and increase their clone size in the same manner thereby adding complexity to the T cell-epitope response. Most importantly, this binding promiscuity ensures broad epitope recognition responses, despite the limited number of unique TCRs within each repertoire".

All changes have been highlighted in the introduction.

Comment 3. In general, I find that the authors tend to mix static description of the repertoire (some clones are large, others are small) with dynamical descriptions, in particular expanded clones. While it is true that a large clone has at some point expanded, because in part of the paper the authors make comparisons between different time points I find it confusing to the use the term expanded to describe a clone that is simply large. This sort of goes together with point 1 as a need to make a clearer distinction between analysis of a snapshot, and analysis that leads to interpretation in terms of dynamics.

Our response: We acknowledge and agree with the comment. Some of the changes go together with changes in response to *comment 1*. We have substituted "expanded clones" with "large clones" to avoid any confusion about monitoring repertoire dynamics. All changes have been highlighted in the main text in yellow.

Comment 4. Here I come to the central issue I have with the paper in its current state, being that there is a lot of qualitative justification for the algorithm, but no quantitative comparison with other algorithms or other options. One first question: for a given value of lambda, varying q ends up being a whole function or the real variable. Even if you only compute it for a finite number of values of q it is dimensionally as large as computing the clone size distribution, which arguably contains as much information as the transformed diversity estimate. So, if one were to cluster, for each value of lambda, clones together in a smart way (and here I have to admit I don't have a definite answer to what this smart would be) and then use instead of the q-based diversity measure the clone size distribution for that value of lambda, it would be much faster to compute. Would the output be less interpretable and predictive? Could such a comparison be made? Is the only issue that it is difficult to cluster clones based on similarity or is there is another why not to go down that road?

Our response: The reviewer raises a very interesting question. Firstly, we agree that there is little quantitative comparisons with other algorithms. We have changed that to include an SVM prediction and Random Forest prediction. We would like to not here that due to low sample size, we couldn't leave one part of the dataset as an external validation. Instead, the model was trained and tested using a leave one out approach. The average accuracy on the train fold prediction was reported. All the folds contained at least one sample for each class. And the same partitioning was employed in both Time and Treatment classification. A more detailed discussion is included on **Pages 13 and 14** and the **Methods section *Classification using SVM and Random Forest***.

Regarding the question on clustering for each value of lambda, we agree that the computation would be much faster. However, as you have pointed out clustering TCRs on similarity is quite difficult at the moment. We have thought about using more sophisticated CDR3 similarity metrics, such as TCRDist or CRD3 dist. They are rooted in conventional approaches and still at their core use substitution matrices which have been developed for sequences from evolutionary related proteins. The balance of cost of computation versus possible information gain weighed in favour of not using these advanced metrics. TCR specificity clustering is a difficult task. More so, clustering repertoires brings in a second layer of information into consideration, thus further complicating the problem. Our rationale for not clustering at the moment, is mainly because of possible information loss that pre clustering could result in. Additionally, figuring out a way to cluster clones in a smart way for each value of lambda would be a project of its own.

Comment 5. I think the idea that sequence similarity leads to closer function should be discussed in a bit more depth, in particular a discussion of the literature and of any quantitative arguments here would be welcome.

Our response:

The idea that sequence similarity leads to a closer function has been derived from ecology, in particular the work of Leinster and Cobbold as shown in reference 42. The importance of sequence similarity in TCR analysis has already been discussed in the introduction. To our knowledge, incorporation of similarity in diversity estimates of T cell repertoires has been performed only by Arora et al [1]. They find that including similarity decreases the overall measure of diversity of a repertoire. This article is already referenced in the manuscript but we have added and highlighted in the main text:

Page 3. *The authors find that "similarity redefines the diversity of complex systems" and it is lower than naive diversity. This indicates that there are cohorts of similar TCRs in the repertoires and it is not excluded that they might have the same biological function e.g. bind similar epitopes.*

Comment 6. "Individual repertoire analysis begins with CDR3 sequence clustering and subsampling in order to reduce computational cost and keep only sequences of interest e.g. such as those that generate functional amino acid sequences." if anything else is removed here, it would be important to mention it, if not it can also be stated clearly.

Our response: We acknowledge the comment and have added further explanations to that sentence:

Page 6. *Individual repertoire analysis begins with CDR_β3 sequence filtering and subsampling in order to reduce computational cost and keep only sequences of interest. The filtering consists of removing all "out-of-frame" sequences and those containing "stop" codons, leaving only "In" frame CDR_β3 sequences. The filtered repertoire is then sampled for non-unique CDR_β3s randomly based on the frequency (count) distribution in the original repertoire in order to preserve the CDR_β3 frequency distribution. The subsampled CDR_β3s are then collapsed to form a list of unique CDR_β3s with their respective counts (frequencies) within the subsampled repertoire.*

Comment 7. When mentioning the "chunks" division in the main text, could you clarify if those are picked at random or structured?

Our response: We acknowledge the comment and have added additional text to clarify the chunk generation:

Page 6. *The "chunks" are formed by dividing the list of all CDR_β3s from the filtered and subsampled repertoire. Each portion of the list ("chunk") is then pairwise compared with the full list of CDR_β3s from the filtered and subsampled repertoire. These portions of the distance matrix contain n rows (n being the chunk length, default value = 100) and S columns (S being the total number of CDR_β3 sequences in the filtered and subsampled repertoire and are outputted in a .tsv format. If concatenated, these portions of the distance matrix form the full CDR_β3 distance matrix and its diagonal is equal to 0.*

Comment 8. Issues regarding sequencing depth are mentioned quite often, and it also written that the diversity profiles you pick overcome some of these issues. I think it would be worth a bit of discussion (describing the issues and how they are overcome) and, if it is already in the literature then just a summary of intuitive reason why similarity scaled diversity helps with these issues.

Our response: We acknowledge the comment. Firstly we have changed the sentence on **Page 7** from "Each dataset was sampled for 50 000 sequences in order to eliminate effects of sequencing depth" to "Each dataset was sampled for 50 000 sequences reduce computational cost and mitigate effects on sequencing depth."

However, we agree that discussion on sequencing depth and how diversity measures play a part in this is very important. Therefore, we have added the following to the Discussion in the main text:

Page 15. *Here, we would also like to emphasise that sequencing depth also influences conventional diversity estimates as absolute, not relative, clone frequencies are needed for diversity estimations [2]. Increasing sequencing depth maximises the number of reads per T cell clone and therefore allows for capturing of a more complete TCR repertoire [2–4]. Experimentally, repertoires from different laboratories or experiments will often be sequenced at a different sequencing depth which makes them difficult to compare using conventional diversity estimates. TCRDivER subsumes most conventional diversity estimates and instead of comparing absolute diversity numbers between repertoires, it focuses on the relationships between these estimates within the repertoire. Therefore, it in some ways bypasses the need for absolute quantitation of TCR frequencies in the repertoire. We are aware that influence of the sequencing depth and experimental sampling still persists in the raw repertoires as the number of unique clones varies between repertoires in both datasets (Supplementary Information Table 1 and 10). Thus we have chosen to subsample the CDR_β3s of the repertoires based on the original frequency distribution and mitigate the influence on the absolute numbers of unique clones (richness) on the diversity estimates. This also normalises the computational cost of each repertoire calculation to under 6 hours, which we find important for practical applications of TCRDivER.*

Comment 9. When you describe picking 50000 sequences at random in the main text and in figure 2 I found it a bit confusing and was not able to understand if you were picking 50000 sequences or 50000 unique sequences. However, I found the explanation in the methods part perfectly clear. Could you try and clarify in the main text and caption of fig2?

Our response: We agree with the comment and have made the following changes:

Fig. 2 caption: *The default subsampling size is 50000 CDR_β3 sequences. Meaning that 50000 non unique CDR_β3s are subsampled and counted to form the filtered and subsampled repertoire with ≤ 50000 unique CDR_β3s with their corresponding counts within the 50000 subsample.*

Page 6. *The filtered repertoire is then sampled for non-unique CDR_β3s randomly based on the frequency (count) distribution in the original repertoire in order to preserve the CDR_β3 frequency distribution. The subsampled CDR_β3s are then collapsed to form a list of unique CDR_β3s with their respective counts (frequencies) within the subsampled repertoire.*

Comment 10. Could you discuss a bit the scale of lambda? I understand that you argue for a range of lambdas, but when picking values to evaluate, how do you decide? What are values of lambda that can be argued to matter in terms of immune function? For instance why do you pick 16 as the highest point before infinity? Maybe your algorithm is a way to find out? I think this discussion would be very interesting for all readers, from clinicians to computer scientists.

Our response:

We agree that the choice of λ s could be systematic. However, due to the time constraints we have not been able to explore this. We feel that the choice is sufficient to capture repertoire biology based on the results of our research. We have added a paragraph to raise awareness that this part of the algorithm hasn't been explored in depth.

Page 19. The range of λ s in this analysis is chosen for higher and lower values separately. For the higher values, the first lambda is chosen as 1, which equals to no similarity scaling. The value of λ is then doubled until $\lambda = 64$, as in a geometric sequence. For lower values $\lambda < 1$ has been incremented by 0.1 for values of 0.1 to 0.5 and 0.25 and 0.75 values were added as well. The users of this algorithm are free to choose their own values by augmenting the configuration file. Given the exploration of mathematics of similarity scaled diversity estimates, we feel that the range captures repertoire information sufficiently. Due to time constraints we have not yet devised an algorithm for choosing the minimum number of *lambda* values needed.

Comment 11. "We observed that for small values of lambda the curves are approximately flat and are evenly spaced on a log scale in all diversity profiles" → link to figure where it is shown would help here

Our response:

We have added a comment referencing Figure 2 of Supplementary information. The change is highlighted in the main text.

Comment 12. Again in the interest of comparison, for repertoire change following immunization, how does your method compare with looking at for instance fold change? There have algorithms developed for that. I understand that the advantage of your method is that you can compare repertoires both longitudinally and between individuals, but it would be important to state for what applications that is essential.

Our response:

We are not sure we fully understand the comment about fold change. We provide a historical account of improvement in estimating repertoire diversity which has in turn improved repertoire classification. We acknowledge that the algorithm should be compared to existing methods and have added a comparison to previous methods (that focus only on naive diversity such as in the work of Greiff et al) using an SVM and Random Forest classifier.

Comment 13. In figure 3B I find the choice of x axis ticks very strange. Why put in a line things that are both ordered in time and ordered by antigen? Why not have several lines then? Or different colors? At the very least not a line plot where there is no clear ordered progression along the x axis.

Our response:

We agree that the choice of the x ticks is not the easiest to understand. However, it was important for us to include untreated profiles in the comparison as well. And they belong to only one timepoint. The choice of ticks is following a progression of timepoints and antigens interchangeably, beginning with timepoint 0 and no antigen stimulation i.e. untreated samples. We welcome further suggestions on how to improve the figure while also including information on untreated repertoires.

Comment 14. In figure 3C what do the colors represent?

Our response: A legend has been added to Fig. 3C.

Comment 15. Regarding hierarchical clustering, you claim that there is very strong separation between different samples. While it is clear that the clusters are much better than random, it is difficult to asses the quality of the clustering without a quantitative point of comparison. Is there any way to think of one?

Our response:

We have thought about this at length, but current methods of comparison of hierarchical clustering are not optimal. We have resorted to following the advice of quantifying separation using an SVM and Random Forest algorithm leaving hierarchical clustering to qualitative interpretation.

Comment 16. Regarding the PCA, you mainly talk about the variance explained by the dimensions, or refer to visual arguments about separation. Could you make the ability of the PCA to separate data quantitative? Is there a way to compare it to a more naive version quantitatively in its ability to predict outcome for human data or separate timepoints or treatment for the mice? For example, I find the analysis of relationship between features very convincing and quite complete. Is there enough data points to try some regression? decision tree? any classification? what about supervised PCA?

Our response: We agree with the reviewers comments about qualitative comparisons. The idea of using PCA in this manner is to use an unsupervised dimensionality reduction technique which one might argue is one of the simplest techniques available. Even with this simplest technique we are able to show that there is an increase in variance as similarity is added to diversity. By using supervised PCA we would run into the same problem of having too little data points to provide an accurate quantitation. We have nonetheless included at the request of the reviewers an SVM and Random Forest classification algorithm. There are obvious shortcomings to this approach due to the number of samples, but we do highlight these for the readers. As these changes are closely tied to previous comments, they are highlighted in the text and we refer the reviewer to our more detailed answer in comment 4.

Comment 17. Link to previous point but you claim that the issue is the lack of data, if so, can you quantify how many repertoires you would need to have a reliable test of predictive power on PD/SD CR/PR groups for instance? In any case I think the claim that data is insufficient to test out the algorithm needs to be discussed a bit more. What about for instance the dataset in ref 38?

Our response:

The dataset in ref 38 [5] is not publicly available anymore in the database. All the information around the study is present but the link to data is broken. This was indeed the study we wanted to handle, as it contains 24 patients at 3 different timepoints. It is difficult to quantify what would be a minimum number of repertoires per class needed to make an accurate classification as this would depend on data quality and classification method.

Please note that with the added references in the revised manuscript the work of Greiff et al is cited with the number 40.

Comment 18. Regarding the choice to reduce the number of sequences to 50 000. Can you explain why that number? The complexity of the algorithm, if I understand correctly depends on the number of clones, not the number of sequences. So, you have to have picked that number based on estimated number of clones. Was it picked for a worst case scenario (i.e clones of size one)? Or based on average number of clones for 50 000 sequences?

Our response:

This is absolutely correct. On our systems we were able to handle up to 50000 clones. That is why we chose this number. We expect that with the rise of computational power this number will rise. Therefore, we have left the choice of number of sequence as a tuneable parameter in TCRDivER.

Comment 19. In section 5.3 you mention that you use these chunks to reduce computational complexity and memory usage. Memory usage seems clear to me here, but I am not sure of how using these chunks reduces complexity. Could you specify? Don't you end up having to compute the whole matrix anyway?

Our response: Storing the chunks instead of a complete distance matrix reduces memory usage and increases speed of computation which is what we aimed to state here. We cannot find the term computational complexity in the main text. The speedup from using chunks doesn't come from the computation of the whole matrix vs chunks. Instead the speedup comes from reading in smaller files (chunks) instead of the full matrix. Our algorithm iterates through the lines of the distance matrix, and keeping the full matrix in memory caused much longer computational times and nodes crashing. We hope that this response clarifies our intention with this sentence.

Comment 20. The paper builds heavily on existing ideas regarding diversity measures, and the papers are generally properly cited. One missing point is that in ref 38 there is already mention of using L'Hôpital's rule to find in particular the link to Shannon's entropy. I think to mention it in the SI.

Our response:

We agree and have added the following to the supplement:

We would like to note that the use of L'Hopitals rule has been established in literature to link scaled diversity measures to Shannon entropy [40]

Please note that with the added references in the revised manuscript the work of Greiff et al is cited with the number 40.

Comment 21. In the SI, given that some calculations are a bit long, it would be good to make it self-standing and define all functions. I think that Z in particular

is not defined (unless I missed it). Numbers for some equations would also make it easier to read in my opinion.

Our response: We thank the reviewer for the comment. We have made all the equations self standing and numbered them accordingly in the SI. Z is defined in equation 7 of main text.

Comment 22. In the last equation of part 1.3, $q=1$, I think it would be clearer to write it.

Our response: We have made the following changes to the SI text:
Rewriting the equation, calculating the limit as $q \rightarrow 1$ and applying L'Hopitals rule:...

Comment 23. In section 1.6, last equation of the series of three, I think the i index near the parenthesis is not necessary given that you are summing already and index is defined by summation right? If I am correct then keeping it makes it confusing.

Our response: We acknowledge the comment and agree. We have removed the index.

Comment 24. You wrote that $\ln(1-x)$ is equivalent to x which is not correct.

Our response: We thank the reviewer for noticing the error. The line in SI has been changed to "By applying the linear approximation $\ln(1-x) \approx -x$ ".

Comment 25. "Note that the final form of the evaluation of $D(q, \lambda)$ $q \rightarrow 0$ is independent of the order of diversity q . It is solely dependent on the distance between CDR3 sequences weighted by their respective frequencies." Does it remain correct when q approaches 0 or infinity?

Our response: We not sure if we understand the comment of the reviewer, but we attempt here to address the comment to the best of our understanding. The original sentence in the SI reads: "Note that the final form of the evaluation of $D(q, \lambda)$ for $\lambda \rightarrow 0$ is independent of the order of diversity q . It is solely dependent on the distance between CDR3 sequences weighted by their respective frequencies." When examining the proof in Section 1.6 Evaluation $\Delta \ln(D(q, \lambda))$ for small λ : Perturbation around $\lambda = 0$ the final form of the equation is devoid of the q parameter. This equation should hold for all qs as it's a perturbation around $\lambda = 0$ of the original similarity scaled diversity.

Comment 26. For the tables showing the profiles in the SI (example is table 14), is there a way to put some of the graphs from different mice, patients or time points (not all) next to each other so that they can be more easily compared? Otherwise, since the reader is not used to this type of graph, I find a bit difficult to compare

Our response: We agree. We've added Figure 4 in the SI, which contains one example repertoire per treatment.

Comment 27. Finally there are a few typos in the text which, once cleared will make reading smoother. A few here: give is written twice in intro, 7 weirdly printed in caption of fig 2, repertoires instead of repertoires a couple of times, in to/into, "is" missing in 5.1, more important some parenthesis typos in the SI that can be misleading

Our response: We acknowledge the the comment and have checked and revised the manuscript for typos and grammar errors. We have also decreased the font size of figure captions to make a more smooth printing and no overlap with page numbers. These have not been highlighted in the text to make sure that the distinction to other comments is preserved.

Response to Reviewer 2:

We thank the reviewers for the comments. We have made changes to the TCRDivER code and to our best ability tried to answer the reviewers comments. We hope that we have done so adequately.

Comment 1. Authors decided to use human TCR sequencing data from cancer patients to validate their findings in immunized - or not - mice repertoires. I understand that the aim was to test the new algorithm rather than focusing on sequencing data, however authors may explain why they used human sequencing data from cancer instead of repertoires from patients affected with autoimmune disorders or generally immune-mediated diseases, that would have matched with immunized murine repertoires more logically. Comments on how this choice may have impacted data analysis could be added in Discussion.

Our response:

We thank the reviewer for the comment and agree with the point. There was a general lack of suitable human data to test our algorithm on. Our initial choice was the dataset described in Greiff et al (in the resubmitted manuscript reference 40). This dataset deals with patients with MS. However, the dataset is no longer publicly available on the database that it was deposited to (the data link is broken). Therefore, we have chosen a dataset that we found the most suitable by number of categories and samples per categories. One argument for this choice was also to show that usage of TCRDivER is not disease or organism dependent.

Comment 2. TCRDivER documentation is not clearly written and hard to follow for a non-computational person. Ideally, 2 pipeline examples should be shown — one using python scripts (local version) and one using bash script (server version). Also, the authors should provide the exact format of input data (what is the column order in tsv file? Does the column names matter?) and a test dataset.

Our response:

We agree with the reviewer and have made significant changes to the code. Due to the computational cost we do not envision our tool to be run on local machines, but have included a note in the readme and provided scripts to run the tool locally. We also included a template data input that needs to be filled out and added an explanation in the readme.

Comment 3. All bash scripts contain a local path “/work3/milvu/” and thus can not be easily run on another server. The github repository contains an explanation how to fix this issue, but it is unclear (see section “specify working directory path - same as folder that is created in the filtering step (I.) on github <https://github.com/sciencisto/TCRDivER>. All bash scripts should take a working directory as a parameter.

Our response:

We agree. We have reorganised the code to be run from a single bash script named TCR-DivER_run.sh. This script takes TCRDivER.config file that takes in parameters for the whole algorithm, including the working directory.

Comment 4. TCRDivER consists of multiple python or bash scripts (sometimes even a bash loop, see “II. Run all distance matrix calculations on <https://github.com/sciencisto/TCRDivER>. The authors should wrap these scripts together in one script or command line tool that takes different arguments as input and not change these arguments in each script directly.

Our response:

We have reorganised the code to be run from a single bash script named TCRDivER_run.sh. This script takes TCRDivER.config file that takes in parameters for the whole algorithm.

Comment 5. On the first step of the TCRDivER pipeline, the authors subsample the number of receptors to 50000. The question is: (1) why does repertoire subsampling not dramatically impact the similarity scaled diversity and (2) why 50000 sequences is enough to represent the diversity? Is the TCRDivER method robust to subsampling, i.e. are two diversity profiles constructed based on two different subsamples of one repertoire similar? Is it correct to compare two diversity profiles constructed based on different number of receptors, e.g. one repertoire contains 50000 receptors and the other only 20000 receptors?

Our response: We acknowledge the comment. We subsample 50000 sequences as this could lead to a maximum of 50000 clones. Our current computational setup allows for this number of clones to be able to run in a reasonable time. The sequences are randomly subsampled based on their frequency, thereby to the best ability preserving the clone distribution. Regarding the comparison of repertoires with uneven number of receptors we have asked ourselves the same. The argument for pursuing this path, was that biologically no two repertoires will have the same number of receptors. This is the basis of usage of richness (clone count) as a diversity index in literature as mentioned in the introduction.

Comment 6. Figure 2: in the caption clone size or frequency is mentioned, however the figure does not show any frequencies on steps I and II. What is “d” on step III?

Our response:

We have added an explanation of d in the figure caption step III, which now reads:

In the graphical representation of the distance matrix d denotes the calculated distance between two pairs of $CDR_{\beta}s$.

We agree. The choice of omitting frequencies was due to the fact that TCRDivER can take in two types of input:

- A list of non-unique CDR3 sequences
- A list of unique CDR3 sequences and their frequencies (or counts)

In case of a list of non-unique CDR3 sequences the algorithm calculates the frequency of each clone. For simplicity of the figure, we have decided to show this input option.

Comment 7. Page 9: “While many of the feature correlation plots are clearly non-linear, they lie on surprisingly tight curves” — any set of points lie on a curve connecting these points. The authors may formalize their statement and ideally perform some sort of numerical analysis to prove it. Also, it is not clear that in Supl11b and Supl11d and thus the validation statement for the murine data is questionable.

Our response:

We thank the reviewer for the comment. We agree the sentence needs to be formalised differently. Supplementary figures 11b and d correspond to the human dataset analysis (in the revised manuscript they are shown in Figure 13). Compared to supplementary figures 13 (in the revised manuscript 15) which correspond to the randomised human dataset, the figures 11b and d still retain structure.

We have formalised the statement differently:

Page 10. *Firstly, while many of the feature correlation plots are clearly non-linear, they lie in a surprisingly tight area.*

Comment 8. Section TCRDivER features improve separation of biologically distinct repertoires. The authors should also compare their method to the original diversity profile methods.

Our response: We agree that this is an important point and thank the reviewer for raising this issue. We have added a comparison of two commonly used diversity indices - Simpson and Shannon diversity index for the murine data in the SI.

As for original diversity profile methods, what we denote as naive diversity is the original diversity profile method for immune stratification. In the introduction we highlight the work of Greiff et al [5] for the use of diversity profiles in attempts of immune stratification. We compare our similarity scaled diversity to naive diversity profiles (no $CDR_{\beta 3}$ distance included), to similarity scaled diversity profiles (which takes in the distance between two $CDR_{\beta 3}$ s as input scaled by the parameter λ), and finally the highlight of our model which is the use of extracted biologically interpretable diversity profile features for stratification. We compare these three methods of assessing diversity by using diversity profiles via PCA analysis and show that there is an additional dimension of separation in the PCA (the total variance is mostly captured by two principal components instead of one). For both of the datasets, including randomised ones this comparison is made in the SI. We also employ SVM and Random Forest classifiers on naive diversity profiles (the same measure as used in the work

of Greiff et al) and similarity scaled diversity profiles. We find that our performance is generally lower, which we believe is due to the group sample sizes as well as the fact that we don't perform any feature prioritisation algorithm. However, similarity scaled diversity outperforms naive diversity with both classifiers when predicting on the treatment category.

Comment 9. The analysis of TCR sequence similarity architecture, diversity and antigen specificity with publicly available TCR datasets has been performed in several recent studies about autoimmune diseases and murine repertoires that may be cited at least somewhere in the Introduction, e.g. 1) [doi:10.7554/eLife.22057](https://doi.org/10.7554/eLife.22057) and 2) <https://doi.org/10.1016/j.ebiom.2021.103429>.

Our response: We have added the references in the introduction Page 2. We feel that Madi et al [6], fit very nicely in the discussion about finding TCR similarity traits, whereas Amorriello et al [7] fit both there, but also in the use of diversity for patient stratification between healthy and diseased donors. The text surrounding the references has been highlighted in yellow.

Comment 10. Page 2 “Entropy, will give give more weight” — typo

Our response: The remark has been acknowledged and the change made accordingly in the revised manuscript.

Comment 11. Page 9: “There are many combinations of features that can be derived from similarity scaled diversity profiles” — which features?

Our response: The sentence has been corrected and it now reads:

Page 10. *There are many combinations of features extracted from diversity profiles that can be derived from similarity scaled diversity profiles.*

Comment 12. Page 10 “This hypothesis seems confirmed when looking at...” — the authors should perform a statistical test or rephrase this sentence.

Our response: We have made the following change to the sentence and the change has been highlighted in the main text accordingly:

Page 11. *And indeed, when looking at the placement of these repertoires along the correlation curve between $\Delta \ln(D(q, \lambda))$ at small λ and the area between profiles for $\lambda = \infty$ and $\lambda = 16$, as shown in the first panel of Fig. 3 C.*

References

1. Arora, R., Burke, H. & Arnaout, R. Immunological Diversity with Similarity. *bioRxiv*, 483131. <https://doi.org/10.1101/483131> (Dec. 2018).
2. Laydon, D. J., Bangham, C. R. & Asquith, B. Estimating T-cell repertoire diversity: Limitations of classical estimators and a new approach. *Philosophical Transactions of the Royal Society B: Biological Sciences* **370**. ISSN: 14712970 (Aug. 2015).
3. Rosati, E. *et al.* Overview of methodologies for T-cell receptor repertoire analysis. *BMC Biotechnology* **17**, 1–16. ISSN: 14726750. <https://bmcbiotechnol.biomedcentral.com/articles/10.1186/s12896-017-0379-9> (July 2017).
4. Calis, J. J. & Rosenberg, B. R. Characterizing immune repertoires by high throughput sequencing: strategies and applications. *Trends Immunol* **35**, 581–590. ISSN: 14714981 (Dec. 2014).
5. Greiff, V. *et al.* A bioinformatic framework for immune repertoire diversity profiling enables detection of immunological status. *Genome Medicine* **7**, 49. ISSN: 1756994X. <http://genomemedicine.com/content/7/1/49> (May 2015).
6. Madi, A. *et al.* T cell receptor repertoires of mice and humans are clustered in similarity networks around conserved public CDR3 sequences. *eLife* **6**. ISSN: 2050084X (July 2017).
7. Amoriello, R. *et al.* TCR repertoire diversity in Multiple Sclerosis: High-dimensional bioinformatics analysis of sequences from brain, cerebrospinal fluid and peripheral blood. *EBioMedicine* **68**. ISSN: 23523964. [http://www.thelancet.com/article/S235239642100222X/fulltext%20http://www.thelancet.com/article/S235239642100222X/abstract%20https://www.thelancet.com/journals/ebiom/article/PIIS2352-3964\(21\)00222-X/abstract](http://www.thelancet.com/article/S235239642100222X/fulltext%20http://www.thelancet.com/article/S235239642100222X/abstract%20https://www.thelancet.com/journals/ebiom/article/PIIS2352-3964(21)00222-X/abstract) (June 2021).

REVIEWERS' COMMENTS:

Reviewer #1 (Remarks to the Author):

The authors have either made changes that answer comments or argued convincingly why they were not needed. Having tested the algorithm on a local machine on a small dataset it still takes a bit of time to adapt it to a local version. It does however work fine and probably much smoother when using the script prepared for the HPC version. I recommend publication, I think this algorithm will be a useful tool for the analysis of TCR repertoires and in particular biomarker development.

Reviewer #2 (Remarks to the Author):

Authors addressed quite well all raised points and concerns in the revised manuscript. I now consider the manuscript suitable for publication.